psychology

resource allocation, inequality aversion, maximin concern, egalitarian concern, mouse tracking, time-series analysis

**Author for correspondence:**
Tatsuya Kameda
e-mail: tkameda@l.u-tokyo.ac.jp

# Reducing variance or helping the poorest? A mouse tracking approach to investigate cognitive bases of inequality aversion in resource allocation

Atsushi Ueshima[1,2] and Tatsuya Kameda[1,3,4]

[1]Department of Social Psychology, The University of Tokyo, 7-3-1 Hongo, Bunkyo-ku, Tokyo 113-0033, Japan
[2]Japan Society for the Promotion of Science, 5-3-1 Kojimachi, Chiyoda-ku, Tokyo 102-0083, Japan
[3]Center for Experimental Research in Social Sciences, Hokkaido University, N10W7 Kita-ku, Sapporo 060-0810, Japan
[4]Brain Science Institute, Tamagawa University, 6-1-1 Tamagawagakuen, Machida, Tokyo 194-0041, Japan

AU, 0000-0002-5850-9526; TK, 0000-0002-6666-493X

Humans dislike unequal allocations. Although often conflated, such 'inequality-averse' preferences are separable into two elements: egalitarian concern about the variance and maximin concern about the poorest (maximizing the minimum). Recent research has shown that the maximin concern operates more robustly in allocation decisions than the egalitarian concern. However, the real-time cognitive dynamics of allocation decisions are still unknown. Here, we examined participants' choice behaviour with high temporal resolution using a mouse-tracking technique. Participants made a series of allocation choices for others between two options: a 'non-Utilitarian option' with both smaller variance and higher minimum pay-off (but a smaller total) compared with the other 'Utilitarian option'. Choice data confirmed that participants had strong inequality-averse preferences, and when choosing non-utilitarian allocations, participants' mouse movements prior to choices were more strongly determined by the minimum elements of the non-Utilitarian options than the variance elements. Furthermore, a time-series analysis revealed that this dominance emerged at a very early stage of decision making (around 500 ms after the stimulus onset), suggesting that the maximin concern operated as a strong cognitive anchor almost instantaneously. Our results provide the first temporally fine-scale evidence that people weigh the maximin concern over the egalitarian concern in distributive judgements.

# 1. Introduction

Former US president Barack Obama characterized the increase of income inequality as the 'defining challenge of our time'. This agenda—'reducing inequality'—is widely supported among the public to unite people in societies. Research across psychology, anthropology and economics has repeatedly shown that humans actually dislike unequal distributions of wealth [1–5]. This 'inequality-averse' preference has been considered to foster human cooperation [6,7] and thus is an essential building block of human societies.

Importantly, recent research has also begun to scrutinize these robust inequality-averse (also called 'inequity-averse') preferences by looking at which element of inequality (variance *per se* or the lowest welfare) tends to be the primary anchor in people's allocation choices [8,9]. These studies have consistently shown that people focus on the worst-off elements of resource distributions more than variance. Although people might often say that they care about equality (in resource distributions) in daily life, what most are actually concerned about is not the variance *per se* but the welfare of the worst-off (i.e. minimum) in resource allocation. In other words, a person's doubt about a socially efficient but unequal distribution may stem not from an egalitarian concern about variance, but rather from a concern for the poorest.

In political domains, it is also critical to note the contrasting implications of the egalitarian concern about variance and the concern for the worst-off in resource allocation. For example, inequality in resource distributions can be reduced by merely depriving the rich of their wealth. However, this in itself does not help the worst-off in society at all [10]. Moreover, in relation to overall efficiency—another key dimension in resource distribution according to utilitarianism [11]—the egalitarian concern about variance is often seen as opposed to pursuing greater overall efficiency, while concern for the worst-off is often politically more compatible with efficiency [12]. Taken together, these studies indicate that egalitarian concern about variance and concern for the worst-off should be considered distinct dimensions of inequality aversion [8–10], although they are often correlated ecologically and thus conflated [9].

Recent experimental research has also revealed that, although people have diverse preferences for resource distribution [13–16], the concern for minima works as a common 'cognitive anchor' across different distributive ideologies [9]. That is, although ideologies (measured using choice preferences for making distributive decisions as a neutral party for others) ranged from 'Rawlsian' (caring most about maximizing the lowest pay-off [17]), to 'egalitarian' (caring about minimizing the variance), to 'utilitarian' (caring most about maximizing the overall amounts [11]) at the behavioural (choice) level, participants universally exhibited the strongest spontaneous attention to minimum pay-offs during information search prior to their choices at the cognitive level [9]. Such a 'maximin concern' (maximizing the minimum) also operated in allocation choices by groups, where participants attended to the minimum—the fate of the least well-off—most closely during conversation toward building a consensus decision as a group [12,18]. These studies indicate that examining cognitive processes during decision making can shed light on the psychological underpinnings of distributive-justice judgements in detail.

Here, we investigate how people may differentiate between the maximin concern and the egalitarian (variance) concern cognitively in a laboratory setting. Using information-search pattern analysis and fMRI analysis, prior work has revealed that minima are weighted more heavily than variances in resource allocation choices [9]. However, these measures have relatively low temporal resolution and do not necessarily reveal the real-time dynamics of cognitive processing during decision making. Using a technique with higher temporal resolution allows us to assess cognitive processes at a finer scale, even in cases where decisions are made very quickly, within a few seconds [19].

To examine the real-time dynamics of cognitive processing during decision making, some recent psychological studies have combined mouse-tracking measures with time-series analysis which takes into account temporal evolution of mouse movement [19,20]. We employed this technique to examine resource allocation choices. Based on previous mouse-tracking studies [21–24], we asked participants to make choices between two allocation options: a Utilitarian option (which is superior in the overall-efficiency dimension) and a non-Utilitarian option (superior in the egalitarian and minimum dimensions)—figure 1a. We recorded each participant's continuous mouse trajectory with high temporal resolution while participants performed the allocation task (see figure 1b for illustration).

We focused on the relative contributions of the maximin parameter and the egalitarian (variance) parameter when participants chose between the Utilitarian and non-Utilitarian options. In figure 1a, for example, the absolute difference in minimum (maximin parameter) is 420 and that in variance (egalitarian parameter) is 0.2. These differences varied from trial to trial. In a trial where the minimum difference is larger between the two options, people are expected to go more directly to the non-Utilitarian option than in a trial where the main difference between the options regards the variance.

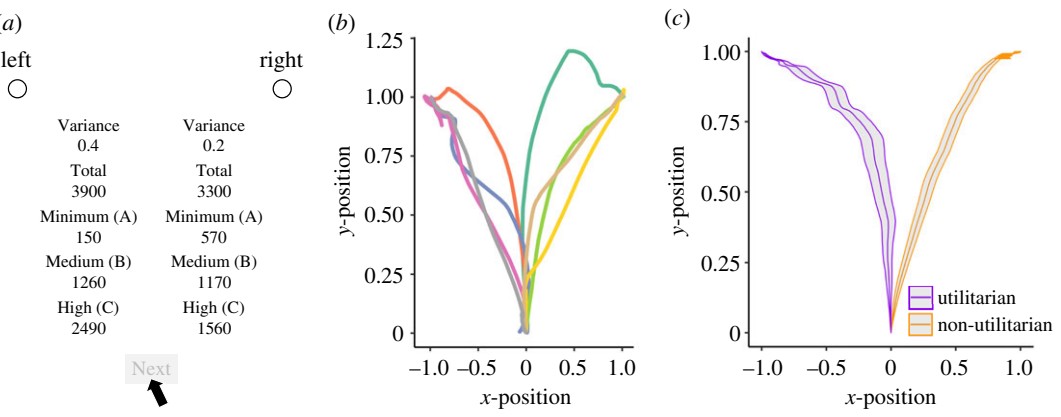

**Figure 1.** Illustrations of a choice problem with two alternatives (the left and right columns) and participants' mouse trajectories. (a) Participants were told that person A would receive the lowest amount (under the 'Minimum' label), person B the middle (Medium), and person C the highest amount (High) in the chosen option (all in Yen). They were also instructed that numbers under the 'Variance' label represented Gini coefficients used in economics that could range from 0 (perfect equality) to 1 (perfect inequality), and those under the 'Total' label indicated the sum of the allocation amounts to the three recipients. In this example, the left is the Utilitarian option, and the right is the non-Utilitarian option. In each trial, the mouse cursor was initially at the 'Next' button, which became clickable after participants had indicated their choice (left or right) using the button located at the top of the screen. (b) An illustration of eight mouse trajectories from one participant. Here, trajectories toward the left represent movements to utilitarian choices and the right represent movements to non-utilitarian choices. The initial location of the mouse cursor on the Next button was coded (0, 0), the coordinate clicked to select the left option was coded (−1, 1), and the coordinate clicked to select the right option was coded (1, 1) in analysis. In the analysis below, the non-utilitarian response was always analysed as being located at $x = 1$ (i.e. the $x$-position of the cursor had a larger positive value as it approached a non-Utilitarian option). (c) Mean trajectories across all participants for utilitarian (purple) and non-utilitarian (orange) choices. Shaded areas indicate a standard error of the mean of $x$-position at each time point (from $t = 1$ to $t = 101$).

We have three hypotheses to test in this study. Firstly, based on the findings that humans have a strong inequality-averse preference [1–5], we predict that:

H1: Non-Utilitarian options will be chosen more frequently than Utilitarian options (as defined above).

Secondly, given that the maximin concern is known to be more robust than the egalitarian (variance) concern in resource allocation [8,9], we predict that:

H2: Participants' choices and overall mouse-tracking patterns will be determined more strongly by the minimum parameter than the egalitarian (variance) parameter of the allocation problems. That is, the worst-off elements will affect behavioural choices as well as cognitive (mouse tracking) processes more robustly, compared with the variance in allocated resources.

To test H2, we compared two models at the behavioural (choice) level. The first model (later described as model A) focuses on the worst-off elements of choice problems (i.e. the difference in minimum value between the two options) to explain participants' choice patterns. The second model (model B) focuses on the variance elements of choice problems (i.e. the difference in variance) to explain participants' choice patterns. As implied in H2, we predicted that model A would provide a better fit to the participants' choices than model B.

At the cognitive level, we predicted that compared with the variance elements, the worst-off elements would exert stronger influence on how straightforwardly participants move the mouse cursor to make choices. As illustrated in figure 1b,c, the cursor's $x$-position was defined to have a larger positive value when the cursor approached the non-Utilitarian options. H2 predicted that the cursor's $x$-position would be determined more strongly by the difference in minimum value than in variance. That is, the larger difference in minimum (rather than in variance) would make the trajectories of the cursor more straightforward to the non-Utilitarian option.

Thirdly, for H3, we introduce a time course analysis of mouse trajectories to shed light on cognitive processes at a finer level. As seen in figure 1b,c, the cursor's $x$-positions did not differ much initially, but

diverged gradually between the two options in each time step. To capture how cognitive focus on the minima over the variances (as claimed in H2) may temporally evolve during decision making, we need to assess the time courses of participants' mouse movements in each time step. Given that spontaneous maximin concern was evinced relatively quickly in both individual and group decision contexts [9,12], we predict that:

H3: Mouse trajectories in earlier stages of decision making, which reflect the psychological dimension that participants focus on sooner to compare merits between the non-Utilitarian and Utilitarian options, will be mainly modulated by the minimum parameter rather than the egalitarian (variance) parameter.

# 2. Material and methods

## 2.1. Participants

Thirty-six students (23 males; mean age = 20.31 ± 1.19) of the University of Tokyo participated in the experiment. Informed consent was obtained from each participant before the experiment using a form approved by the Institutional Review Board of the Graduate School of Humanities and Sociology at the University of Tokyo.

## 2.2. Task

Our allocation task consisted of 48 choice problems in total (for a full list of the choice sets, see electronic supplementary material, table S1). Participants were provided two options in each trial (see the left and right columns in figure 1a). Participants chose one of two options as a neutral party for three anonymous others (labelled as person A, B and C), who were participants from another experiment. Participants were told that person A would receive the lowest (numbers under the 'Minimum' label), person B the middle (Medium), and person C the highest amount (High) in the chosen option; and that, after the experiment, the result of one choice problem would be randomly drawn from the 48 problems to determine the amounts paid to the recipients. In addition to Minimum, Medium and High amounts of allocation, we also provided two quantitative summaries of each option: the Gini coefficient of resource distribution (Variance) and the sum of the allocation amounts (Total). In each trial, one option always had a larger Total than the other option. In other words, the former option was superior in terms of the overall efficiency (Utilitarian) dimension, whereas the latter was superior in terms of the egalitarian (variance) as well as the maximin (minimum) dimension. Hereafter, the former is called the 'Utilitarian' option (e.g. the left option in figure 1a), and the latter is called the 'non-Utilitarian' option (e.g. the right option). The presentation order (i.e. left or right) of each option was randomized across participants. The presentation order of the 48 problems was randomized for each participant. Participants were also informed that they would remain completely anonymous to the recipients. The recipients were paid according to the allocations after the experiment.

Choice problems were presented on a 21.5-inch monitor with 1920 × 1080 pixel resolution. In each trial, the mouse cursor was initially at the 'Next' button located at the bottom centre of the screen (figure 1a). Participants indicated their choice by moving the mouse cursor upward to click either option's check button located at the top right or left. Each trial ended when participants clicked the Next button, which became clickable after they had indicated their choice in the trial. By this procedure, every trial started with the cursor positioned on the Next button. We adopted this protocol from Mathur & Reichling [25].

## 2.3. Models for behavioural choices about allocation

To test whether participants' behavioural choices were determined more strongly by the minimum parameter than the egalitarian (variance) parameter (i.e. the behavioural part of H2), we conducted a series of model analyses. Here, we adopted two utility models ('quasi-maximin model' [13] and 'mean-variance model' [26]), both of which have been used in economics and finance. We used Bayes factor [27] for model comparison, which quantified the evidence for model A over model B.

In the following, we designated the 'quasi-maximin model' [9,13] as model A. This model assumes that choice behaviour can be approximated as a trade-off between the minimum and total parameters.

Denoting the pay-offs allocated to the three recipients as $\pi_1$, $\pi_2$, $\pi_3$, the utility of option $x$ for participant $i$ is defined as

$$U_i(x) = \alpha_i \times \min[\pi_1, \pi_2, \pi_3] + (1 - \alpha_i) \times (\pi_1 + \pi_2 + \pi_3), \qquad (2.1)$$

where $\alpha_i \in [0, 1]$ indicates the degree of the participant's concern for the minimum parameter. We used a non-informative prior distribution for $\alpha$ parameter (see electronic supplementary material for the detailed model specification).

We designated the 'mean-variance model' [26] as model B. This model assumes that choice behaviour can be approximated as a trade-off between the variance and total parameters. That is, the utility of option $x$ for participant $i$ is defined as

$$U_i(x) = \frac{1}{3}(\pi_1 + \pi_2 + \pi_3) - \beta_i \times \text{variance}[\pi_1, \pi_2, \pi_3], \qquad (2.2)$$

where $\beta_i$ is raw-scale and unconstrained (i.e. it ranges from negative infinity to positive infinity) and indicates the degree of the participant's concern for the egalitarian (variance) parameter. A smaller $\beta$ indicates that a participant prefers Utilitarian options more, and a larger $\beta$ indicates that a participant prefers non-Utilitarian options more. We used an informative prior distribution for $\beta$ parameter (see electronic supplementary material for the detailed model specification). However, results reported below were unchanged under different prior specifications.

## 2.4. Mouse trajectories prior to making choices

The cursor's position $(x, y)$ during decision making was sampled at 60 Hz, using a JavaScript program made by Mathur & Reichling [25]. All experimental procedures were carried out using Qualtrics (https://www.qualtrics.com). In accordance with prior work [19], the initial location of the mouse cursor on the Next button was coded $(0, 0)$, the coordinate clicked to select the left option was coded $(-1, 1)$ and the coordinate clicked to select the right option was coded $(1, 1)$ in analysis. Figure 1$c$ illustrates mean cursor trajectories across all participants, with $x = 1.0$ indicating the choice of a non-Utilitarian option.

## 2.5. Resampling

To time-normalize all participants' responses across trials, each trial's cursor trajectory was interpolated to represent the same length. Following the procedure recommended by Spivey *et al.* [23], we sliced the duration of each trial into 101 equal-sized time bins. The position $(0, 0)$ was defined as $t = 1$, and the position at which participants clicked the option's check box was defined as $t = 101$.

## 2.6. Analysis of the mouse trajectories

To see which of the two parameters (minimum or variance) determined participants' cognitive processes more strongly, we analysed each participant's cursor trajectory during decision making in each trial. Because our hypotheses are only relevant to the right or left position of the cursor until making choices (figure 1$a$), we focus on the cursor's $x$-position in the following analyses.

To examine the cognitive dynamics of allocation choices within a trial, it is necessary to examine the influence of the minimum parameter and the egalitarian (variance) parameter on the cursor's position at every time point (from $t = 1$ to $t = 101$ as defined earlier). For this purpose, for each of the 48 problems, we first calculated the absolute differences in both minimum and variance between the Utilitarian and non-Utilitarian options. For example, in the example shown in figure 1$a$, the absolute difference in minimum is 420 and that in variance is 0.2. The time-series analysis with these values as predictors allows us to verify which parameter, minimum or variance, contributes more to the mouse cursor's position during decision making.

Here, we used a state-space model (equation (2.3)). In cases where the non-Utilitarian option was shown at the left and the Utilitarian option at the right of the screen, the cursor's $x$-position was inverted in analysis so that it always took a positive value toward the non-Utilitarian option. Our state-space model took into account the autocorrelation between data points at $t$ and $t - 1$ by

assuming that the observed cursor's position was sampled from a latent state. The cursor's position $x$ at time $t$ by participant $i$ in trial $j$ is given by the following equations:

$$x_{t,i,j} = \mu_{t,i} + \boldsymbol{\beta}_{\mathrm{Min}_{t,i}} \times \mathrm{Diff\_min}_j + \boldsymbol{\beta}_{\mathrm{Var}_{t,i}} \times \mathrm{Diff\_var}_j + \varepsilon,$$

$$\mu_{t,i} = \mu_{t-1,i} + \delta_i,$$

$$\boldsymbol{\beta}_{\mathrm{Min}_{t,i}} = \boldsymbol{\beta}_{\mathrm{Min\_population}_t} + \boldsymbol{\eta}_{\mathrm{Min}_t},$$

$$\boldsymbol{\beta}_{\mathrm{Var}_{t,i}} = \boldsymbol{\beta}_{\mathrm{Var\_population}_t} + \boldsymbol{\eta}_{\mathrm{Var}_t},$$

$$\boldsymbol{\beta}_{\mathrm{Min\_population}_t} = \boldsymbol{\beta}_{\mathrm{Min\_population}_{t-1}} + \zeta_{\mathrm{Min}},$$

$$\boldsymbol{\beta}_{\mathrm{Var\_population}_t} = \boldsymbol{\beta}_{\mathrm{Var\_population}_{t-1}} + \zeta_{\mathrm{Var}},$$

where $\varepsilon \sim N(0, 0.005)$, $\delta_i \sim N(0, \sigma_{\delta_i})$, $\boldsymbol{\eta}_{\mathrm{Min}_t} \sim N(0, \sigma_{\eta_{\mathrm{Min}_t}})$, $\boldsymbol{\eta}_{Var_t} \sim N(0, \sigma_{\eta_{\mathrm{Var}_t}})$,

$$\zeta_{\mathrm{Min}} \sim N(0, \sigma_{\zeta_{\mathrm{Min}}}) \text{ and } \zeta_{\mathrm{Var}} \sim N(0, \sigma_{\zeta_{\mathrm{Var}}}). \tag{2.3}$$

We quantified the population-level influence of each parameter on the cursor's $x$-position at time $t$ with $\boldsymbol{\beta}_{\mathrm{Min\_population}_t}$ and $\boldsymbol{\beta}_{\mathrm{Var\_population}_t}$, which evolved over time. If the 95% pointwise credible interval of $\boldsymbol{\beta}_{\mathrm{Min\_population}_t}$ did not include zero, it is interpreted with this model that the cursor's $x$-position at time $t$ was credibly modulated by the difference in minimum between the two options. At $t = 1$, the cursor's position was always defined as $(0, 0)$, so the coefficient was zero. For H2 and H3, we investigated whether $\boldsymbol{\beta}_{\mathrm{Min\_population}_t}$ was larger than $\boldsymbol{\beta}_{\mathrm{Var\_population}_t}$ as a general trend (H2), and which of these two coefficients departed from zero earlier (i.e. engaged participants' cognitive focus sooner) during decision making (H3).

To take into account individual heterogeneity, we used a hierarchical model in equation (2.3). The influence of the difference between minima on participant $i$'s cursor position at time $t$ was denoted as $\boldsymbol{\beta}_{\mathrm{Min}_{t,i}}$, which was generated from a normal distribution with mean $\boldsymbol{\beta}_{\mathrm{Min\_population}_t}$ and standard deviation $\boldsymbol{\eta}_{\mathrm{Min}_t}$ (the same was the case with the variance parameter $\boldsymbol{\beta}_{\mathrm{Var}_{t,i}}$). For more detailed descriptions, see electronic supplementary material, table S2. The explanatory variable $\mathrm{Diff\_min}_j$ indicated the absolute difference in minimum, the variable $\mathrm{Diff\_var}_j$ the absolute difference in Gini coefficients between the Utilitarian and Non-Utilitarian options in choice problem $j$ $(1, \ldots, 48)$. These variables were standardized with mean zero and variance one so that the magnitude of $\boldsymbol{\beta}_{\mathrm{Min\_population}_t}$ and $\boldsymbol{\beta}_{\mathrm{Var\_population}_t}$ could be compared. The Pearson correlation coefficient between Diff_min and Diff_var across the 48 choice problems was 0.49.

## 2.7. Estimation

In the analysis, we used Markov chain Monte Carlo (MCMC) methods for parameter estimation. All the models we used in the current study, including models A and B, the logistic regression (to be mentioned later), and the state-space model, were implemented using rstan v. 2.19.3 [28] and brms v. 2.12.0 [29] with R v. 3.6.3 [30]. Bayes factor was calculated with bridgesampling v. 1.0–0 [31]. We used the $\hat{R}$ statistic (the Gelman–Rubin convergence statistic) to check for convergence of all models' parameter estimations [32]. The $\hat{R}$ statistics were below 1.1 for all the parameters we estimated using MCMC, indicating the convergence of our MCMC simulations. A correlation analysis was implemented with JASP 0.12.2 [33]. In all analyses including the mouse-tracking analysis, we used all the data from each trial, whether participants chose the utilitarian or the non-Utilitarian option in the trial.

# 3. Results

## 3.1. Choice data

We analysed the choice data for the 48 allocation problems. As shown in figure 2$a$, non-Utilitarian options were chosen more frequently than Utilitarian options ($\boldsymbol{\beta}_{\mathrm{intercept}} = 1.87$; [0.91, 2.90] 95% credible interval: a mixed-effects logistic regression). This confirms H1, replicating the results of previous research that participants were generally inequality averse in resource allocations [1–3,6].

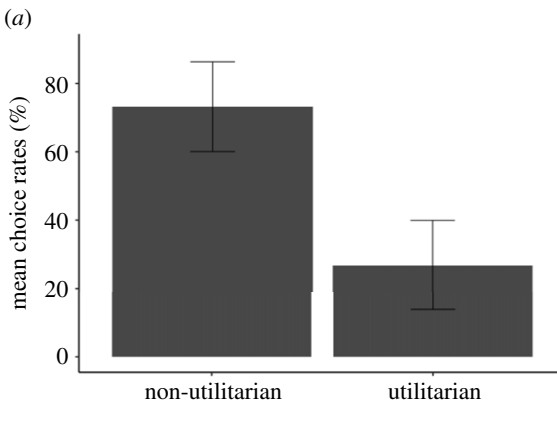
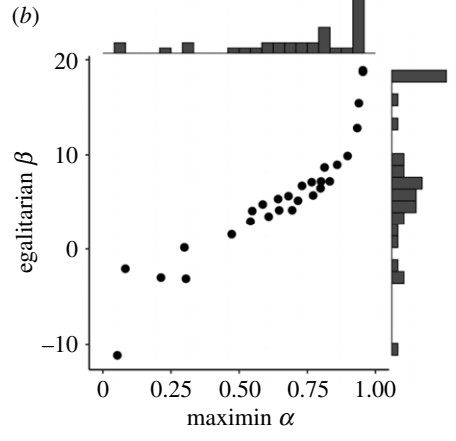

**Figure 2.** Results of behavioural data. (a) Proportions of choices for each option (Utilitarian or non-Utilitarian). Error bars represent 95% confidence intervals. (b) Distributions of participants' maximin $\alpha$ and egalitarian $\beta$ estimated for the two utility models (see the text). Note that some data points overlap, but the density for each dot can be seen from the histograms shown at the upper and right sides.

## 3.2. Model-based choice analysis

First, we report basic data from the model-based analysis. Figure 2b shows the distributions of each participant's 'maximin $\alpha$' obtained using model A (equation (2.1)) and 'egalitarian $\beta$' obtained using model B (equation (2.2)). The histograms shown in the upper and right sides of figure 2b indicate that both $\alpha$ and $\beta$ had relatively large values, reflecting participants' generic inequality-averse preferences. Not surprisingly, $\alpha$ and $\beta$ were correlated, $r = 0.91$ (95% CI [0.81, 0.95]).

However, these two models should be different in terms of plausibility. According to H2, we predicted that model A should be better than model B at fitting participants' behavioural choices. Calculated Bayes factor was $\mathrm{BF}_{AB} = 65.26$, which means that the observed choice behaviour was 65 times more likely to have occurred under model A than under model B [34]. This result suggests that participants' choices were determined more strongly by the minimum parameter than the variance parameter, confirming the behavioural part of H2. For more detailed model descriptions (including the specification of prior distributions, sensitivity analyses under different prior distributions, and different model specifications for the mean-variance model and the quasi-maximin model), see the electronic supplementary material.

## 3.3. Mouse tracking

Before analysing the mouse trajectory data, we present response time data to clarify the time scale of mouse-tracking analysis. Figure 3a displays a distribution of participants' response times that elapsed from the trial onset to their clicking one of the choice buttons. As seen in the figure, the average response time was rather short, with mean = 1.97 s and s.d. = 1.07 s. We also conducted an analysis to see if there are any meaningful differences in response time between utilitarian choices and non-utilitarian choices. Mean response time was 2.12 s (s.d. = 1.13) for utilitarian responses and 1.91 s (s.d. = 1.04) for non-utilitarian responses. A linear mixed model indicated that there were no meaningful differences in response time between utilitarian and non-utilitarian choices ($\boldsymbol{\beta}_{\text{non-Utilitarian choice}} = -0.01$; 95% CI [−0.13, 0.11]: a mixed-effects linear regression).

## 3.4. Observed relative contributions of the minimum parameter and the variance parameter to mouse tracking.

To test the cognitive part of H2, we compared the magnitude of weights of the minimum parameter ($\boldsymbol{\beta}_{\text{Min\_population}_t}$) and that of the egalitarian (variance) parameter ($\boldsymbol{\beta}_{\text{Var\_population}_t}$). Figure 3b displays the time trajectory of the two weights. As expected, the overall mouse trajectories indicated that the minimum parameter was weighted more strongly than the egalitarian (variance) parameter, at almost all time points except from $t = 1$ to $t = 24$ and from $t = 50$ to $t = 69$ (for more detail, see electronic supplementary material, table S3).

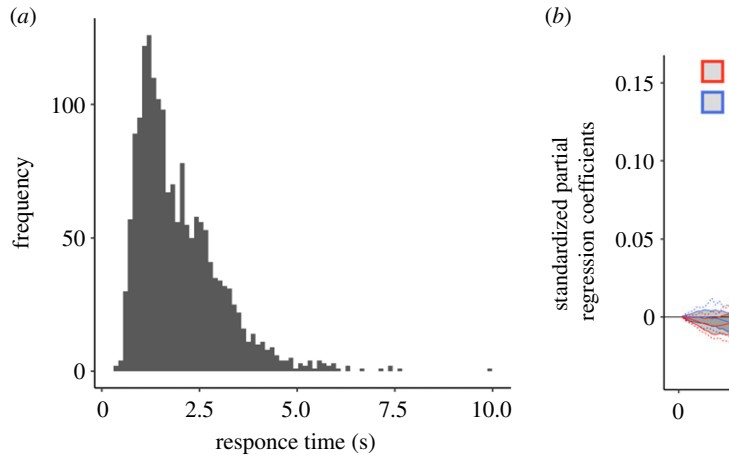
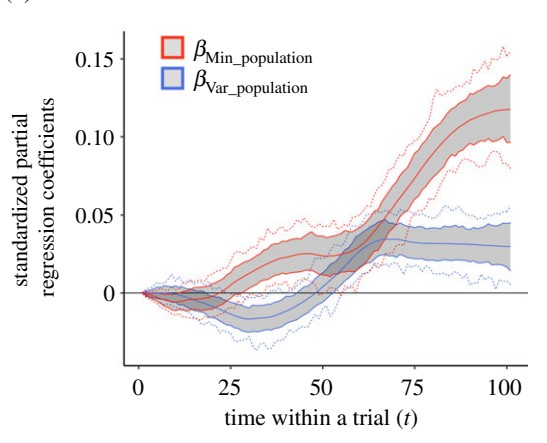

**Figure 3.** Results of time-series analysis on mouse trajectory during decision making. (*a*) Distribution of response time collapsed across participants and the 48 problems. Decision responses were rather quick in each trial, with mean = 1.97 s and s.d. = 1.07 s. (*b*) Trajectories of weights of the minimum parameter ($\beta_{\text{Min\_population}_t}$) and weights of the variance parameter ($\beta_{\text{Var\_population}_t}$) to determine the location of the mouse cursor, from $t = 1$ (onset of the trial) to $t = 101$ (when clicking the option's check box). Here, $\beta_{\text{Min\_population}_t}$ quantified the extent to which the absolute differences in minima predicted mouse movement toward the non-utilitarian option at time $t$. Around $t = 1$ to $t = 30$, $\beta_{\text{Min\_population}_t}$ was not meaningfully different from zero (a horizontal line). Shaded areas indicate 95% pointwise highest density interval of estimated coefficients. Dotted lines indicate 95% simultaneous highest density interval of estimated coefficients.

The time-series model also revealed that the weights of the minimum parameter ($\boldsymbol{\beta}_{\text{Min\_population}_t}$) credibly departed from zero at an earlier stage of decision making (from $t = 29$), while that of the egalitarian (variance) parameter ($\boldsymbol{\beta}_{\text{Var\_population}_t}$) remained around or below zero until a middle stage ($t = 54$). These patterns confirm H3's prediction that participants would focus on the minimum parameter sooner than the egalitarian (variance) parameter. These results provide the first evidence from real-time cognitive dynamics that the maximin concern plays a more important role than the egalitarian (variance) concern in resource allocation choices. We also conducted an additional analysis by statistically controlling for the total element in the mouse-tracking analysis and confirmed that results were essentially unchanged. See the electronic supplementary material (Mouse-tracking analysis after controlling for the total element) and electronic supplementary material, table S4.

## 4. Discussion

Social distribution of resources is a fundamental process in all human societies. In distributive choices, the egalitarian concern about variance and the maximin concern about the worst-off are often conflated. This conflation itself may be an inevitable social phenomenon, because the minimum and variance parameters of resource distributions are often ecologically correlated [9]. However, these two concerns can have drastically different political and economic implications, and thus should be treated as different dimensions underlying people's robust inequality-averse preferences. The present study focused on the cognitive operation of these two concerns during a third-party resource allocation task. We first confirmed the robust inequality-averse preferences (H1), where non-Utilitarian options were chosen more frequently (73.2% on average) than Utilitarian options (26.8% on average). The model comparison also revealed that participants' inequity-averse preferences were better approximated by the quasi-maximin model (equation (2.1)) than by the mean-variance model (equation (2.2)). This means that the inequity-averse preferences are particularly strong when the difference in minima (rather than variance) is large between choice options. Secondly, the mouse-tracking technique combined with a time-series model revealed that, overall, the difference in minima affected participants' mouse trajectory during decision making more strongly than the difference in variances (H2). Moreover, the same model also confirmed that the minimum (but not the variance) started to influence the cursor's trajectory at a very early stage of decision making (around 500 ms after the stimulus onset), indicating that the dimension that participants focused on sooner was the minimum parameter rather than the egalitarian (variance) parameter (H3).

The main advantage of our mouse-tracking method in this study, compared with previous studies, is that it can reveal real-time cognitive dynamics by analysing the movement of the cursor with high temporal resolution [19,21]. We have shown that the mouse-tracking approach is a promising step toward building cognitive models of distributive decisions. Furthermore, in contrast to most previous studies that provided the maximin (maximizing the minimum amount) and the egalitarian (minimizing the variance) allocations as two distinct options from the outset [8,15], participants in our study were only asked to make choices between Utilitarian and non-Utilitarian options. Thus, it was left completely up to the participants on which element (if any) of the non-Utilitarian options (variance or minimum) to focus. In this sense, our less-intrusive method provides a stricter test bed for the thesis that people voluntarily prioritize the minimum dimension over the variance dimension in resource allocation decisions.

There are several limitations in this study. Firstly, we still do not understand the psychological processes underlying how and why the minimum dimension is prioritized more than the egalitarian dimension. Although we observed that such selective focus was instantiated voluntarily from a very early stage of decision making, this result does not necessarily imply that computations regarding the minimum parameter rely on the so-called 'intuitive processes' [35]. In fact, a prior study has indicated that maximin preferences may be less intuitive compared with egalitarian preferences [36]. Moreover, the neural mechanism related to the maximin concern in allocation choices has been shown to mainly involve the right temporoparietal junction (rTPJ) [9], which has been associated with higher cognitive and social functions such as perspective taking [37–41]. Future research should address how and why the minimum parameter operates spontaneously from an early stage of decision making and whether deliberative processes may further facilitate or hinder its operation in resource allocation choices [18].

Secondly, we did not assess individual differences in this study, such as Big Five personality traits, or emotional or cognitive empathy [42]. An interesting future direction will be to combine these psychological measures with the mouse-tracking approach to shed light on how individual differences may affect cognitive dynamics during allocation choices.

Thirdly, we did not conduct manipulation checks to make sure that participants actually believed that their decisions would affect monetary allocation to real people. We did not conduct manipulation checks because asking such questions might cause ungrounded suspicion among participants that we use deception in our experiments. However, the lack of belief checks remains as a limitation of the current study.

Fourthly, in this study, there was a moderate correlation ($r = 0.49$) between the maximin and variance parameters. As described in the note of electronic supplementary material, table S1, we generated the 48 choice sets to systematically manipulate utilitarian (i.e. total) and non-utilitarian (i.e. minimum and variance) parameters. As argued elsewhere [9], a moderate correlation between the maximin and variance parameters often characterizes everyday choice settings. Arguably, keeping such a moderate correlation in the laboratory may contribute to understanding people's ordinary choices in ecologically natural settings [43]. On the other hand, we admit that the current design could have affected the statistical estimation of the independent effects of the maximin and variance parameters because of possible collinearity ('aliasing'; see [44]). For example, we observed that around $t = 30$–$40$, a higher variance parameter predicted mouse movement toward the Utilitarian option. This pattern is difficult to interpret but may have arisen spuriously due to aliasing. Future research that strikes a better balance between ecological considerations and statistical concerns will be important toward fuller understanding of the cognitive mechanisms underpinning allocation choices.

# 5. Conclusion

Recent studies have examined which element of inequality-averse preference (variance *per se* or the lowest welfare) plays an important role in resource allocation choices [8,9]. Building on these studies, the present study investigated how the egalitarian concern and the maximin concern affect decision making using a mouse-tracking technique. We replicated prior findings that people are generally inequality averse and confirmed that the minimum parameter played a more important role than the variance parameter in such choices. Critically, we provided the first evidence with high temporal resolution data that the distinction between the maximin and the egalitarian (variance) concerns was voluntarily initiated from a very early stage of decision making. These results clearly show that people differentiate the maximin concern and the egalitarian concern not only at the behavioural (choice) level but also at the cognitive level. Although these two concerns are apparently similar and

often confounded in daily settings, our study indicated that the minimum and the egalitarian dimensions are distinguishable in people's thinking about distributive-justice judgements.

Ethics. This study was approved by the Institutional Review Board of the Graduate School of Humanities and Sociology at the University of Tokyo. Informed consent was obtained from each participant before the experiment.

Data accessibility. All data are available at: https://figshare.com/s/c501dc92b15a13452b66. The 48 choice sets used in the experiment have been uploaded as part of the electronic supplementary material.

Authors' contributions. A.U. and T.K. developed the study concept. A.U. contributed to the study design. Data collection and analysis were performed by A.U. The manuscript was written and approved by A.U. and T.K.

Competing interests. We have no competing interests.

Funding. This research was supported by the Japan Society for the Promotion of Science Grant-in-Aid for Scientific Research JP16H06324, and Japan Science and Technology Agency CREST grant no. JPMJCR17A4 (17941861) to T.K., and Japan Society for the Promotion of Science (JSPS) Grant-in-Aid for JSPS fellows JP18J21498 to A.U. Support from CiSHub at the University of Tokyo is also appreciated.

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
