## [Peer Review File · Royal Society Open Science]

Review History

RSOS-201159.R0 (Original submission)

Review form: Reviewer 1 (Scott Tindale)

Is the manuscript scientifically sound in its present form?

Yes

Are the interpretations and conclusions justified by the results?

Yes

Is the language acceptable?

Yes

Do you have any ethical concerns with this paper?

No

Have you any concerns about statistical analyses in this paper?

No

Recommendation?

Accept with minor revision (please list in comments)

Comments to the Author(s)

I think the paper presents interesting results on an important topic using innovative techniques (the mouse tracking). I don't really have any issues with what they have done, but thought that some ancillary analyses might be useful. For example, did the mouse tracking data differ depending on whether they chose the utilitarian or non-utilitarian response? Were there time differences for the difference responses or were there characteristics of the particular problems that led to more or less utilitarian responses or was it more a function of individual differences? Some additional analyses may help to illuminate more about the specific cognitive processes that underlie such judgments. It just seems there is more here to be discovered.

Review form: Reviewer 2

Is the manuscript scientifically sound in its present form?

No

Are the interpretations and conclusions justified by the results?

No

Is the language acceptable?

Yes

Do you have any ethical concerns with this paper?

Yes

Have you any concerns about statistical analyses in this paper?

Yes

Recommendation?

Major revision is needed (please make suggestions in comments)

Comments to the Author(s)

Please see above.

Review form: Reviewer 3

Is the manuscript scientifically sound in its present form?

No

Are the interpretations and conclusions justified by the results?

Yes

Is the language acceptable?

Yes

Do you have any ethical concerns with this paper?

No

Have you any concerns about statistical analyses in this paper?

No

Recommendation?

Accept with minor revision (please list in comments)

Comments to the Author(s)

As specified to the editor, I am not a specialist in Social Psychology, so my comments below regard specifically the methodology and analysis used in this manuscript, and the ways in which they are presented. I believe the paper has a lot of potential, but it can (and should) be improved before publication. Below, I specified a number of aspects that could be improved, by order of importance.

1. The authors are not explicit enough about how their hypotheses are expected to be assessed in the data. This is particularly problematic for hypothesis 2 and 3, which established that the worse-off elements (minimum) should influence more strongly participants' choices than variance in allocated resources. From the moment in which these hypotheses are discussed (p.7), the reader should have an intuitive understanding of what exactly should they expect to see in mouse trajectories. This is not provided in the manuscript. As a result, it becomes hard to understand exactly why are they using the mouse-tracking methodology (as opposed to other time-sensitive measures, such as response times!) as well as why are they performing the analysis they do later on (time-step model). There are a number of possible things the authors could do to solve this issue. I mention two of them:

a. The linking hypothesis between the hypothesised cognitive process (i.e. the variables that play a role in decision) and the mouse-trajectories should be made explicit. For example, the authors might expect that how fast or how straightforwardly participants make their non-utilitarian decision is determined by the difference in minimum values rather than by the difference in variance. If this were the case, one might even be able to notice a visual difference when comparing by-participant mean trajectories for trials where the difference between the two minima values is maximal and cases in which this difference is minimal. The predictions, stated in these lines, should be part of the paper.

b. The inclusion of figures showing the actual mouse tracking results might help the reader. For example, Figure 1b as it is is not informative, as it doesn't have information about which trajectories correspond to utilitarian or non-utilitarian choices. Having a colour coding-scheme would be optimal. Similarly, it would be useful to have a figure illustrating the mean trajectories (or just the x-trajectory) for each kind of choice. That is, roughly illustrating the raw properties of mouse trajectories which are later used for the model.

2. In relation with the first point, I find the description of the time-series analysis performed on mouse trajectories not fully transparent. I think the authors should provide a more complete explanation of what each of the terms is doing, and how exactly are we supposed to interpret the figure 3b. This could be done in the Supplementary Materials, but note that, as it is, the formula in page 13 are not useful, as it's unclear exactly how they are extracted and what exactly each term means (for instance, is there a difference between η min and η _min?)

3. The authors need to clarify that what is meant by "decision time" is the time when participants click on the response button (i.e. response time). This is not obvious in MT literature, where

people use the method precisely to determine the exact moment in which participants make the decision, which happens before the actual click.

4. The description of the design of the experiment is rather confusing. The authors present the picture as if they are manipulating whether the choices are utilitarian vs. Not-utilitarian, but I think they could make clearer how these options relate to the manipulation of Total, Variance and Minimum (they do an attempt of this I p.88 lines 143-148, but it's not explicitly presented as actual factors they are manipulating).

5. What is figure 3a trying to show? Wouldn't be more pertinent to have an idea about whether the manipulation has some effect on decision time? (See also my point above about the meaning of decision time)

6. It's not always clear whether the analysis of mouse-tracking data is being made only for trials where the final choice was non-utilitarian or for all trials. I guess it's always the former, but clarification is needed.

Additional Comments about statistics

- In section "Estimation" (p.9, line 174), it's unclear what model are they referring to when they talk about "the model", as well as what is the R^2 statistics they have run... (is it a correlation?). I believe the authors can choose to be vague in the paper about the exact statistics they did if and only if they provide all the details in the supplementary materials, which is not the case. One should be able to run the analyses on their data, and understand exactly what they did.

- Bayes factors are generally reported together with some measure of their robustness under different prior specifications. This is lacking (as well as any reference to which prior specification was used).

Decision letter (RSOS-201159.R0)

Dear Dr Kameda

The Editors assigned to your paper RSOS-201159 "Reducing variance or helping the poorest? A mouse tracking approach to investigate cognitive bases of inequality aversion in resource allocation" have now received comments from reviewers and would like you to revise the paper in accordance with the reviewer comments and any comments from the Editors. Please note this decision does not guarantee eventual acceptance.

We do not generally allow multiple rounds of revision so we urge you to make every effort to fully address all of the comments at this stage. If deemed necessary by the Editors, your

manuscript will be sent back to one or more of the original reviewers for assessment. If the original reviewers are not available, we may invite new reviewers.

Please submit your revised manuscript and required files (see below) no later than 21 days from today's (ie 13-Oct-2020) date. Note: the ScholarOne system will 'lock' if submission of the revision is attempted 21 or more days after the deadline. If you do not think you will be able to meet this deadline please contact the editorial office immediately.

Kind regards,

Anita Kristiansen
Editorial Coordinator

on behalf of Essi Viding (Subject Editor)
openscience@royalsociety.org

Associate Editor Comments to Author:
Comments to the Author:

Thank you for the submission, and please accept our apologies for the unusual delay in completing review - this is largely a function of the COVID pandemic reducing the availability of potential referees, and the journal is doubly grateful, therefore, for the support of the reviewers who have commented.

Given the comments of the reviewers, you have a range of queries and comments to respond to before we can consider the paper further. Please carefully review and respond to the queries - making clear any changes made in the revised paper, and supplying a point-by-point response to the referees. Note that we are not generally able to offer multiple rounds of revision, so please work hard to address the reviewers' concerns.

Reviewer comments to Author:
Reviewer: 1

Comments to the Author(s)

I think the paper presents interesting results on an important topic using innovative techniques (the mouse tracking). I don't really have any issues with what they have done, but thought that some ancillary analyses might be useful. For example, did the mouse tracking data differ depending on whether they chose the utilitarian or non-utilitarian response? Were there time differences for the difference responses or were there characteristics of the particular problems

that led to more or less utilitarian responses or was it more a function of individual differences? Some additional analyses may help to illuminate more about the specific cognitive processes that underlie such judgments. It just seems there is more here to be discovered.

Reviewer: 2

Comments to the Author(s)

OVERALL COMMENTS

This manuscript investigates an interesting question regarding moral intuitions and seeks to shed light on these intuitive judgments through a sensible application of real-time mouse tracking. I will comment primarily on methodology as I am not an expert in moral psychology. The mouse-tracking work appears to have been carried out well, but I do have a number of important concerns about the experimental design, particularly the confounding of manipulated parameters with one another and the model specifications, as detailed below. I believe it is critical for

MAJOR COMMENTS

- 1.) I am concerned that the authors' efforts to estimate the separate contributions of the maximin and egalitarian parameters to choosing the utilitarian vs. non-utilitarian option may be seriously compromised by the use of an experimental design that manipulated these parameters non-independently. The correlation between the maximin and egalitarian parameters was 0.49, and from Table S1, it appears that even the parameters of the utilitarian option might have been manipulated non-independently from the parameters of the non-utilitarian option. Depending on exactly how these various parameters were varied jointly, this kind of experimental design can potentially lead to "aliasing", a serious statistical problem in which one cannot parse the effects of the various manipulations from one another (i.e., aliasing of main effects) or cannot parse the effects of the manipulations from interactions among them (i.e., aliasing of main effects with interactions). It is critical that the paper provide more information on how and why the parameters were manipulated as they were. How were the choice sets generated, what was the resolution of the experimental design, and what does this imply about aliasing and our ability to actually draw conclusions about the independent effects of the maximin and egalitarian parameters, and of the utilitarian vs. non-utilitarian options? (See reference [1] for an excellent overview on aliasing and resolution. The R package AlgDesign may be helpful as well.)
- 2.) The specifications for both Models A and B (supplement) use highly informative priors on the population average parameters (alpha and beta) that favor strong preferences for maximin and egalitarian choices. Are the Bayes factor of 65 and the general results robust to noninformative prior choices?
- 3.) In Model B, why is the coefficient on the utilitarian component equal to 1/3 instead of $(1 - \beta_i)$? With the current specification, isn't it the case that a higher β_i reduces the utility of ALL options? If so, doesn't this affect estimation of the population average in a way that is a function of the arbitrary 1/3 coefficient?
- 4.) Were any precautions taken, or manipulation checks conducted, to make sure subjects actually believed that their decisions would affect money allocation to real people? I worry that social desirability bias could overwhelm subjects' genuine moral intuitions if they correctly recognize that their allocations are fictional.

MINOR COMMENTS

- 5.) The Introduction should more clearly distinguish between contributions of resource allocation parameters to actual choices versus to cognitive dynamics before the choice is made.

- 6.) Page 7, H2: Unclear what is meant by “parametric analysis”. This is a vast class of statistical methods.
- 7.) The Methods section confused with me with regard to statistical methods, for example referring to MCMC methods for “parameter estimation” before I had learned what the parameters of interest and models were. The later material showing the models and notation is good; please move it to Methods.
- 8.) Page 12: “Parametric absolute differences” sounds like it refers to the estimates of a parametric model, but are you not just referring to differences between the fixed design parameters of the experiment?
- 9.) In the time-series analysis, are the confidence intervals simultaneous or pointwise? If pointwise, please also provide simultaneous CIs in the figures.
- 10.) Top of page 10: What is the scale of the estimated beta? Odds?
- 11.) Figure 3b: Around $t=30-40$, the coefficients for maximin and variance options are in different directions. Does this mean that during this time period, a higher variance parameter predicted mouse movement toward the *utilitarian* option? Why might this be? If I am interpreting this counterintuitive finding correctly, note that this is the kind of finding that could potentially arise spuriously due to aliasing.
- 12.) Figure 1B: Hard to interpret this when the points aren't connected. Maybe take a random sample of the trajectories and plot them as lines in different colors.
- 13.) The FigShare link does not seem to work.

TYPOS

- 13.) Page 5: “reveal real the time”

REFERENCES

- 1.) Collins LM, Dziak JJ, Li R. Design of experiments with multiple independent variables: a resource management perspective on complete and reduced factorial designs. *Psychological Methods*. 2009 Sep;14(3):202.

Thank you for the opportunity to review this manuscript.

Reviewer: 3

Comments to the Author(s)

As specified to the editor, I am not a specialist in Social Psychology, so my comments below regard specifically the methodology and analysis used in this manuscript, and the ways in which they are presented. I believe the paper has a lot of potential, but it can (and should) be improved before publication. Below, I specified a number of aspects that could be improved, by order of importance.

1. The authors are not explicit enough about how their hypotheses are expected to be assessed in the data. This is particularly problematic for hypothesis 2 and 3, which established that the worse-off elements (minimum) should influence more strongly participants' choices than

variance in allocated resources. From the moment in which these hypotheses are discussed (p.7), the reader should have an intuitive understanding of what exactly should they expect to see in mouse trajectories. This is not provided in the manuscript. As a result, it becomes hard to understand exactly why are they using the mouse-tracking methodology (as opposed to other time-sensitive measures, such as response times!) as well as why are they performing the analysis they do later on (time-step model). There are a number of possible things the authors could do to solve this issue. I mention two of them:

a. The linking hypothesis between the hypothesised cognitive process (i.e. the variables that play a role in decision) and the mouse-trajectories should be made explicit. For example, the authors might expect that how fast or how straightforwardly participants make their non-utilitarian decision is determined by the difference in minimum values rather than by the difference in variance. If this were the case, one might even be able to notice a visual difference when comparing by-participant mean trajectories for trials where the difference between the two minima values is maximal and cases in which this difference is minimal. The predictions, stated in these lines, should be part of the paper.

b. The inclusion of figures showing the actual mouse tracking results might help the reader. For example, Figure 1b as it is is not informative, as it doesn't have information about which trajectories correspond to utilitarian or non-utilitarian choices. Having a colour coding-scheme would be optimal. Similarly, it would be useful to have a figure illustrating the mean trajectories (or just the x-trajectory) for each kind of choice. That is, roughly illustrating the raw properties of mouse trajectories which are later used for the model.

2. In relation with the first point, I find the description of the time-series analysis performed on mouse trajectories not fully transparent. I think the authors should provide a more complete explanation of what each of the terms is doing, and how exactly are we supposed to interpret the figure 3b. This could be done in the Supplementary Materials, but note that, as it is, the formula in page 13 are not useful, as it's unclear exactly how they are extracted and what exactly each term means (for instance, is there a difference between η_{\min} and η_{\min} ?)

3. The authors need to clarify that what is meant by "decision time" is the time when participants click on the response button (i.e. response time). This is not obvious in MT literature, where people use the method precisely to determine the exact moment in which participants make the decision, which happens before the actual click.

4. The description of the design of the experiment is rather confusing. The authors present the picture as if they are manipulating whether the choices are utilitarian vs. Not-utilitarian, but I think they could make clearer how these options relate to the manipulation of Total, Variance and Minimum (they do an attempt of this I p.88 lines 143-148, but it's not explicitly presented as actual factors they are manipulating).

5. What is figure 3a trying to show? Wouldn't be more pertinent to have an idea about whether the manipulation has some effect on decision time? (See also my point above about the meaning of decision time)

6. It's not always clear whether the analysis of mouse-tracking data is being made only for trials where the final choice was non-utilitarian or for all trials. I guess it's always the former, but clarification is needed.

Additional Comments about statistics

- In section "Estimation" (p.9, line 174), it's unclear what model are they referring to when they talk about "the model", as well as what is the R^2 statistics they have run... (is it a correlation?). I

believe the authors can chose to be vague in the paper about the exact statistics they did if and only if they provide all the details in the supplementary materials, which is not the case. One should be able to run the analyses on their data, and understand exactly what they did.

- Bayes factors are generally reported together with some measure of their robustness under different prior specifications. This is lacking (as well as any reference to which prior specification was used).

===PREPARING YOUR MANUSCRIPT===

===PREPARING YOUR REVISION IN SCHOLARONE===

Author's Response to Decision Letter for (RSOS-201159.R0)

See Appendix A.

RSOS-201159.R1 (Revision)

Review form: Reviewer 2

Is the manuscript scientifically sound in its present form?

No

Are the interpretations and conclusions justified by the results?

No

Is the language acceptable?

Yes

Do you have any ethical concerns with this paper?

No

Have you any concerns about statistical analyses in this paper?

Yes

Recommendation?

Major revision is needed (please make suggestions in comments)

Comments to the Author(s)

The authors have addressed some of my comments, though unfortunately some of my major concerns remain, in particular #1 below. As I describe below, this is a fundamental problem for the interpretation of the main analyses, and I worry that half of the paper's main conclusions may not be justified as a result. I hope this can be addressed more carefully using revised analyses, as I mention below.

1.) I appreciate the authors' clarification of how the choice sets were designed, but this structure of choice sets is a more serious problem for the central conclusions of the manuscript than the authors' revision suggests.

It is fine to manipulate parameters in an aliased manner in order to address an ecological hypothesis, as the revision implies, but then one simply cannot rigorously address hypotheses such as H2, one of the paper's two central hypotheses ("...choices and mouse trajectories by the minimum parameter than by the egalitarian (variance) parameter"), at least with the current analysis approach. Accordingly, central conclusions appearing throughout the Abstract and main text, such as:

"Our results provide the first...evidence that people weight the maximin concern over the egalitarian concern..."

could simply be spurious. It could simply be that other (confounded) aspects of the choice sets, such as the total, is driving the apparent evidence in favor of subjects' weighting maximin concerns over egalitarian concerns.

There are statistical methods to address this analytically, e.g., by doing subset analyses in which the variables of interest (e.g., minimum and variance) are not confounded by other manipulated elements (e.g., the total) or by statistically controlling for the other manipulated elements. These are methods that control for confounding, because aliasing is a type of confounding. I would strongly encourage the authors to look into how they could resolve this problem rigorously by

considering the specific structure and resolution of the choice sets. It may be as simple as revising all models to include all manipulated choice set parameters rather than just the ones of interest (thus controlling for the other, confounded parameters).

2.) The sensitivity analyses regarding the prior on beta are a good, and reassuring, addition. Thank you.

However, with alpha, you state that its $U(0,1)$ is noninformative, but can't alpha take on negative values as well, at least in principle? Bounding alpha above 0 excludes the possibility that any subject wants the minimum to be *smaller* rather than larger. We don't expect that to be the case on average, but this prior entirely excludes this possibility for every individual subject, which would seem to be a rather strong assumption. It is rarely advisable to use priors that completely exclude some parts of the parameter space. I would advise performing a sensitivity analysis using a genuinely noninformative or highly diffuse prior, similar to what was done for beta in the revision.

3.) Excellent. Thank you.

4.) My concern was about subject's beliefs about the veracity of their allocations, not about the veracity of the allocations themselves. I would expect at least some mention of this limitation, given that no manipulation checks were performed. (Minor point: regarding the actual veracity of the allocation, the old and revised manuscript just say "The participants were paid after the experiment" at the indicated line; you might revise this indicate that they were actually paid according to the allocations).

My minor comments were addressed adequately.

Review form: Reviewer 3

Is the manuscript scientifically sound in its present form?

Yes

Are the interpretations and conclusions justified by the results?

Yes

Is the language acceptable?

Yes

Do you have any ethical concerns with this paper?

No

Have you any concerns about statistical analyses in this paper?

No

Recommendation?

Accept with minor revision (please list in comments)

Comments to the Author(s)

I think the authors have significantly improved the manuscript and addressed most of my concerns. I consider the manuscript should be accepted for publication. Still, I encourage them to make some minor changes, which I specify below.

1. One of my suggestions in my previous review was to include a figure showing mean mouse tracking trajectories per type of response. In particular, I think it would be very helpful to see mean trajectories across subjects which include some by-participant measure of variability (e.g., SE on participant-means). In the new Fig.1c, the authors include mean trajectories but only for one participant. This decision is rather odd to me (i.e., how did they choose what participant to show), and it doesn't really target my original concern, which was having some figure that shows how much variability is in there there (between subjects variability). Note that Fig1b is already there and does a great job in showing *_within_* subject variability. I think the authors should replace Fig1c with a new figure showing mean trajectories across subjects.
2. The caption of Fig1 can also be improved. For example, each of three subcaptions should be unified (right now there are two a/b/c).
3. In the caption of figure 1b, the authors say that "In the analysis below, we reset the x-position of the cursor...". If I understand correctly, what the authors mean by this is that the non-utilitarian answer will always be analysed as being located at $x=1$. If this is the case, this is not clear from their description.
4. At the point in which H2 is introduced (lines 110-114), it's still unclear for the reader how the manipulation of the minimum parameter and the egalitarian variance is done. I think it will be helpful to, using the example in Fig.1a, to clearly explain that in a trial where the minimum difference is larger between the two options, people are expected to go more straightforward to the non-utilitarian option than in a trial where the main difference between the options regards the variance.

Decision letter (RSOS-201159.R1)

Dear Dr Kameda

The Editors assigned to your paper RSOS-201159.R1 "Reducing variance or helping the poorest? A mouse tracking approach to investigate cognitive bases of inequality aversion in resource allocation" have now received comments from reviewers and would like you to revise the paper in accordance with the reviewer comments and any comments from the Editors. Please note this decision does not guarantee eventual acceptance.

Please submit your revised manuscript and required files (see below) no later than 21 days from today's (ie 04-Jan-2021) date. Note: the ScholarOne system will 'lock' if submission of the revision is attempted 21 or more days after the deadline. If you do not think you will be able to meet this deadline please contact the editorial office immediately.

Kind regards,

Andrew Dunn

on behalf of Prof Essi Viding (Subject Editor)

Associate Editor Comments to Author:

You will note that of the three reviewers who have now seen your paper, two have recommended the paper be accepted. However, one of the reviewers has consistently recommended major revisions - indeed, they note that you have not fully addressed their concerns from the initial round of review. Under most circumstances, we would reject your paper at this stage, but because you have taken some steps to get the paper to a publishable standard, we are prepared to give you the benefit of the doubt: you have this final opportunity to revise the paper and address the concerns raised by the critical reviewer here. Please be aware that if this reviewer is not satisfied with your revisions, we will be forced to reject your paper, so do ensure you carefully and clearly respond to their criticisms.

Reviewer comments to Author:

Reviewer: 2

Comments to the Author(s)

The authors have addressed some of my comments, though unfortunately some of my major concerns remain, in particular #1 below. As I describe below, this is a fundamental problem for the interpretation of the main analyses, and I worry that half of the paper's main conclusions may not be justified as a result. I hope this can be addressed more carefully using revised analyses, as I mention below.

1.) I appreciate the authors' clarification of how the choice sets were designed, but this structure of choice sets is a more serious problem for the central conclusions of the manuscript than the authors' revision suggests.

It is fine to manipulate parameters in an aliased manner in order to address an ecological hypothesis, as the revision implies, but then one simply cannot rigorously address hypotheses such as H2, one of the paper's two central hypotheses ("...choices and mouse trajectories by the minimum parameter than by the egalitarian (variance) parameter"), at least with the current analysis approach. Accordingly, central conclusions appearing throughout the Abstract and main text, such as:

“Our results provide the first...evidence that people weight the maximin concern over the egalitarian concern...”

could simply be spurious. It could simply be that other (confounded) aspects of the choice sets, such as the total, is driving the apparent evidence in favor of subjects’ weighting maximin concerns over egalitarian concerns.

There are statistical methods to address this analytically, e.g., by doing subset analyses in which the variables of interest (e.g., minimum and variance) are not confounded by other manipulated elements (e.g., the total) or by statistically controlling for the other manipulated elements. These are methods that control for confounding, because aliasing is a type of confounding. I would strongly encourage the authors to look into how they could resolve this problem rigorously by considering the specific structure and resolution of the choice sets. It may be as simple as revising all models to include all manipulated choice set parameters rather than just the ones of interest (thus controlling for the other, confounded parameters).

2.) The sensitivity analyses regarding the prior on beta are a good, and reassuring, addition. Thank you.

However, with alpha, you state that its $U(0,1)$ is noninformative, but can’t alpha take on negative values as well, at least in principle? Bounding alpha above 0 excludes the possibility that any subject wants the minimum to be *smaller* rather than larger. We don’t expect that to be the case on average, but this prior entirely excludes this possibility for every individual subject, which would seem to be a rather strong assumption. It is rarely advisable to use priors that completely exclude some parts of the parameter space. I would advise performing a sensitivity analysis using a genuinely noninformative or highly diffuse prior, similar to what was done for beta in the revision.

3.) Excellent. Thank you.

4.) My concern was about subject’s beliefs about the veracity of their allocations, not about the veracity of the allocations themselves. I would expect at least some mention of this limitation, given that no manipulation checks were performed. (Minor point: regarding the actual veracity of the allocation, the old and revised manuscript just say “The participants were paid after the experiment” at the indicated line; you might revise this indicate that they were actually paid according to the allocations).

My minor comments were addressed adequately.

Reviewer: 3

Comments to the Author(s)

I think the authors have significantly improved the manuscript and addressed most of my concerns. I consider the manuscript should be accepted for publication. Still, I encourage them to make some minor changes, which I specify below.

1. One of my suggestions in my previous review was to include a figure showing mean mouse tracking trajectories per type of response. In particular, I think it would be very helpful to see mean trajectories across subjects which include some by-participant measure of variability (e.g., SE on participant-means). In the new Fig.1c, the authors include mean trajectories but only for one participant. This decision is rather odd to me (i.e., how did they choose what participant to show), and it doesn't really target my original concern, which was having some figure that shows

how much variability is in there there (between subjects variability). Note that Fig1b is already there and does a great job in showing *_within_* subject variability. I think the authors should replace Fig1c with a new figure showing mean trajectories across subjects.

2. The caption of Fig1 can also be improved. For example, each of three subcaptions should be unified (right now there are two a/b/c).

3. In the caption of figure 1b, the authors say that "In the analysis below, we reset the x-position of the cursor...". If I understand correctly, what the authors mean by this is that the non-utilitarian answer will always be analysed as being located at $x=1$. If this is the case, this is not clear from their description.

4. At the point in which H2 is introduced (lines 110-114), it's still unclear for the reader how the manipulation of the minimum parameter and the egalitarian variance is done. I think it will be helpful to, using the example in Fig.1a, to clearly explain that in a trial where the minimum difference is larger between the two options, people are expected to go more straightforward to the non-utilitarian option than in a trial where the main difference between the options regards the variance.

===PREPARING YOUR MANUSCRIPT===

===PREPARING YOUR REVISION IN SCHOLARONE===

To revise your manuscript, log into <https://mc.manuscriptcentral.com/rsos> and enter your Author Centre - this may be accessed by clicking on "Author" in the dark toolbar at the top of the

page (just below the journal name). You will find your manuscript listed under "Manuscripts with Decisions". Under "Actions", click on "Create a Revision".

Author's Response to Decision Letter for (RSOS-201159.R1)

See Appendix B.

RSOS-201159.R2 (Revision)

Review form: Reviewer 2

Is the manuscript scientifically sound in its present form?

Yes

Are the interpretations and conclusions justified by the results?

Yes

Is the language acceptable?

Yes

Do you have any ethical concerns with this paper?

No

Have you any concerns about statistical analyses in this paper?

No

Recommendation?

Accept as is

Comments to the Author(s)

I thank the authors for responded thoroughly to my comments. I have no further concerns and am happy to recommend acceptance.

Decision letter (RSOS-201159.R2)

Dear Dr Kameda,

It is a pleasure to accept your manuscript entitled "Reducing variance or helping the poorest? A mouse tracking approach to investigate cognitive bases of inequality aversion in resource allocation" in its current form for publication in Royal Society Open Science.

You can expect to receive a proof of your article in the near future. Please contact the editorial office (openscience@royalsociety.org) and the production office (openscience_proofs@royalsociety.org) to let us know if you are likely to be away from e-mail

contact – if you are going to be away, please nominate a co-author (if available) to manage the proofing process, and ensure they are copied into your email to the journal.

on behalf of Prof Essi Viding (Subject Editor)
openscience@royalsociety.org

Reviewer comments to Author:
Reviewer: 2

Comments to the Author(s)
I thank the authors for responded thoroughly to my comments. I have no further concerns and am happy to recommend acceptance.

Appendix A

October 31, 2020

Dear Dr. Essi Viding:

Thank you for your and your reviewers' thoughtful comments on our manuscript "Reducing variance or helping the poorest? A mouse tracking approach to investigate cognitive bases of inequality aversion in resource allocation" submitted for publication in *Royal Society Open Science* (RSOS-201159). We have taken these comments into careful consideration and revised our manuscript to reflect them. Below we summarize how we have responded to each of the points raised by the two reviewers. The experiment, figure, table, and line numbers refer to those in the revised manuscript unless otherwise specified.

Responses to the comments of Reviewer #1

(1) I think the paper presents interesting results on an important topic using innovative techniques (the mouse tracking). I don't really have any issues with what they have done, but thought that some ancillary analyses might be useful. For example, did the mouse tracking data differ depending on whether they chose the utilitarian or non-utilitarian response? Were there time differences for the difference responses or were there characteristics of the particular problems that led to more or less utilitarian responses or was it more a function of individual differences? Some additional analyses may help to illuminate more about the specific cognitive processes that underlie such judgments. It just seems there is more here to be discovered.

Response

Thank you for raising these important points. We think that there are two points to be addressed.

The first point regards differences in mouse tracking or response time data between utilitarian and non-utilitarian responses. As for the mouse tracking data, using a time-series model, we had shown in the old manuscript that the key parameters (i.e., minimum and variance) of a choice set determined participants' cursor trajectories prior to making choices at a very fine scale (Fig

3b). The choice results (the utilitarian or non-utilitarian responses) have already been included in this analysis, and we think that we do not have much to add here. As for response time differences for the different choice responses (i.e., utilitarian or non-utilitarian), we have conducted an additional analysis to see if there were any meaningful differences in response time between the utilitarian choices and non-utilitarian choices. Mean response time was 2.12 s (SD = 1.13) for utilitarian responses and 1.91 s (SD = 1.04) for non-utilitarian responses. A linear mixed model indicated that there were no meaningful differences in response time between utilitarian and non-utilitarian choices ($\beta_{non-utilitarian\ choice} = -0.01$; 95% CI [-0.13, 0.11]; a mixed-effects linear regression). We have reported these results in the revised manuscript as follows:

Revised manuscript (lines 308-313):

“We also conducted an analysis to see if there are any meaningful differences in response time between utilitarian choices and non-utilitarian choices. Mean response time was 2.12 s (SD = 1.13) for utilitarian responses and 1.91 s (SD = 1.04) for non-utilitarian responses. A linear mixed model indicated that there were no meaningful differences in response time between utilitarian and non-utilitarian choices ($\beta_{non-utilitarian\ choice} = -0.01$; 95% CI [-0.13, 0.11]; a mixed-effects linear regression).”

The second point is about whether the particular choice problems or individual differences led to more or less utilitarian responses. We did not assess individual differences in terms of psychological variables (such as empathy [Davis, 1983] and Big Five personality traits) in the present study. We have discussed this limitation as follows:

Revised manuscript (lines 378-382):

“we did not assess individual differences in this study, such as Big Five personality traits, or emotional or cognitive empathy [42]. An interesting future direction will be to combine these psychological measures with the mouse-tracking approach to shed light on how individual differences may affect cognitive dynamics during allocation choices.”

As for whether the particular choice problems led to more or less utilitarian responses, we believe that this point has already been addressed by our model-based choice analysis with models A (quasi-maximin model) and B (mean-variance model). We have clarified this as follows:

Revised manuscript (lines 342-346):

“The model comparison also revealed that participants’ inequity-averse preferences were better approximated by the quasi-maximin model (Eq. 1) than by the mean-variance model (Eq. 2). This means that the inequity-averse preferences are particularly strong when the difference in minimums (rather than variance) is large between choice options.”

References

Davis, M. H. (1983). Measuring individual differences in empathy: Evidence for a multidimensional approach. *Journal of Personality and Social Psychology*, 44(1), 113-126.

Responses to the comments of Reviewer #2

MAJOR COMMENTS

(1) *I am concerned that the authors' efforts to estimate the separate contributions of the maximin and egalitarian parameters to choosing the utilitarian vs. non-utilitarian option may be seriously compromised by the use of an experimental design that manipulated these parameters non-independently. The correlation between the maximin and egalitarian parameters was 0.49, and from Table S1, it appears that even the parameters of the utilitarian option might have been manipulated non-independently from the parameters of the non-utilitarian option. Depending on exactly how these various parameters were varied jointly, this kind of experimental design can potentially lead to "aliasing", a serious statistical problem in which one cannot parse the effects of the various manipulations from one another (i.e., aliasing of main effects) or cannot parse the effects of the manipulations from interactions among them (i.e., aliasing of main effects with interactions). It is critical that the paper provide more information on how and why the parameters were manipulated as they were. How were the choice sets generated, what was the resolution of the experimental design, and what does this imply about aliasing and our ability to actually draw conclusions about the independent effects of the maximin and egalitarian parameters, and of the utilitarian vs. non-utilitarian options? (See reference [1] for an excellent overview on aliasing and resolution. The R package AlgDesign may be helpful as well.)*

Response

Thank you for raising this important point. First, we regret the lack of information about how the maximin and egalitarian parameters and 48 choice sets were generated in the old manuscript. We have added a description to the note of Table S1 about how the 48 choice sets were constructed:

Revised manuscript:

"These 48 choice sets were constructed to distinguish two types of distributive concerns in choices [i.e., utilitarian (larger total) and non-utilitarian (smaller variance and larger minimum)]. Out of the 48 choice sets, the egalitarian and utilitarian parameters are orthogonal to each other in 24 choice sets (i.e., the Pearson correlation coefficient between absolute differences in variance and that in total is zero for these 24 choice sets), and the maximin and utilitarian parameters are orthogonal to each other in the remaining 24 choice sets (i.e., the

Pearson correlation coefficient between absolute differences in minimum and that in total is zero).”

We have also acknowledged that caution is required to interpret the results from the current experimental design due to the possible collinearity between the maximin and variance parameters, as pointed out by the reviewer:

Revised manuscript (lines 383-396):

“Thirdly, in this study, there was a moderate correlation ($r = 0.49$) between the maximin and variance parameters. As described in the note of Table S1, we generated the 48 choice sets to systematically manipulate utilitarian (i.e., total) and non-utilitarian (i.e., minimum and variance) parameters. As argued elsewhere [9], a moderate correlation between the maximin and variance parameters often characterises everyday choice settings. Arguably, keeping such a moderate correlation in the laboratory may contribute to understanding people’s ordinary choices in ecologically natural settings [43]. On the other hand, we admit that the current design could have affected the statistical estimation of the independent effects of the maximin and variance parameters because of possible collinearity (“aliasing”; see [44]). For example, we observed that around $t = 30 \sim 40$, a higher variance parameter predicted mouse movement toward the utilitarian option. This pattern is difficult to interpret but may have arisen spuriously due to aliasing. Future research that strikes a better balance between ecological considerations and statistical concerns will be important toward fuller understanding of the cognitive mechanisms underpinning allocation choices.”

(2) The specifications for both Models A and B (supplement) use highly informative priors on the population average parameters (alpha and beta) that favor strong preferences for maximin and egalitarian choices. Are the Bayes factor of 65 and the general results robust to noninformative prior choices?

Response

We regret the vague information about the prior distributions of both models A (Eq. 1) and B (Eq. 2). Denoting the money allocated to the three recipients as π_1 ,

π_2, π_3 , the utility of option x for participant i in model A (quasi-maximin model) is given by:

$$U_i(x) = \alpha_i \times \min[\pi_1, \pi_2, \pi_3] + (1 - \alpha_i) \times (\pi_1 + \pi_2 + \pi_3),$$

$$\alpha_i = \alpha_{population} + \alpha_{difference_i},$$

$$\alpha_{population} \sim \text{Uniform}(0, 1),$$

$$\alpha_{difference_i} \sim \text{Normal}(0, \sigma_\alpha),$$

$$\sigma_\alpha \sim \text{Cauchy}(0, 5)$$

Here, the population average parameters $\alpha_{population}$ range from 0 to 1 (please see red colored notation above). If $\alpha = 0$, participants make completely utilitarian choices. If $\alpha = 1$, participants make completely non-utilitarian choices (see Figure 2b for each participant's α estimated from data). Here, a uniform distribution between 0 and 1 was used for the prior of α . That is, we did not use an informative prior for α in the old manuscript.

Next, model B is given by:

$$U_i(x) = \frac{1}{3}(\pi_1 + \pi_2 + \pi_3) - \beta_i \times \text{variance}[\pi_1, \pi_2, \pi_3],$$

$$\beta_i = \beta_{population} + \beta_{difference_i},$$

$$\beta_{population} \sim \text{Normal}(5, 15),$$

$$\beta_{difference_i} \sim \text{Normal}(0, \sigma_\beta),$$

$$\sigma_\beta \sim \text{Cauchy}(0, 5)$$

As the reviewer pointed out, we used an informative prior for $\beta_{population}$ (a normal distribution with mean = 5 and standard deviation = 15, please see red colored notation above). Actually, we had conducted analyses with three

different priors for β and reported the most conservative result as the Bayes factor in favor of model A, in which model B provided the best fit.

The following table provides the details. The first column corresponds to the results reported in the main text. The analysis in the second column used a normal distribution of mean = 0 and standard deviation = 15 for the prior distribution of β , which assumed no particular preferences for maximin and egalitarian choice (i.e., non-utilitarian choice) because the prior distribution of mean equals 0. The analysis in the third column used a normal distribution of mean = 0 and standard deviation = 100, which again assumed no particular preference for maximin and egalitarian choice (i.e., mean = 0) but with less informative (i.e., standard deviation = 100) prior distribution.

	Mean = 5, sd = 15 (reported in the main manuscript)	Mean = 0, sd = 15	Mean = 0, sd = 100
Bayes factor (model A vs. B)	65.26	65.67	319.58

We also confirmed that making the prior distribution flatter (e.g., sd = 1000) yields an even larger Bayes factor.

We kept the same prior distribution for model A as in the old manuscript, because this prior distribution setting is non-informative as we explained above.

The figures below show the correlations between β estimated under the prior distribution reported in the old manuscript and β estimated under the two new prior distributions explained above.

New egalitarian β 's prior specification:
Mean = 0, SD = 15 (normal distribution)

New egalitarian β 's prior specification:
Mean = 0, SD = 100 (normal distribution)

β values are highly correlated, showing that the estimated β values are robust across different prior specifications.

We have reported these sensitivity analyses in the supplement and added a description of these analyses in the revised manuscript as follows:

Revised manuscript (lines 300-302):

“For more detailed model descriptions (including the specification of prior distributions, sensitivity analyses under different prior distributions, and a different model specification for mean-variance model), see the electronic supplementary material.”

(3) In Model B, why is the coefficient on the utilitarian component equal to 1/3 instead of (1-beta_i)? With the current specification, isn't it the case that a higher beta_i reduces the utility of ALL options? If so, doesn't this affect estimation of the population average in a way that is a function of the arbitrary 1/3 coefficient?

Response

Setting the coefficient on the utilitarian component equal to 1/3 in Model B follows the tradition of the “mean-variance” utility model (Markowitz, 1952), which has been used as a standard model in economics and finance. The previous research from which this study derived (Kameda et al., 2016) also used

this specification for the purpose of model comparison. It is true that, as the reviewer pointed out, a higher β_i reduces the utility of all options with this specification. However, this does not affect our conclusions, because our model comparison did not hinge on the absolute value of the population average β . That is, we estimated Bayes factor between the two models and also the correlation between α and β , which are not affected by the absolute value of β .

To verify this point, we have reset the coefficient on the utilitarian component as $(1-\beta_i)$ and conducted an additional analysis (model B-2) as follows:

$$U_i(x) = (1 - \beta_i)(\pi_1 + \pi_2 + \pi_3) + \beta_i \times (-\text{variance}[\pi_1, \pi_2, \pi_3]),$$

$$\beta_i = \beta_{\text{population}} + \beta_{\text{difference}_i},$$

$$\beta_{\text{population}} \sim \text{Uniform}(0, 1),$$

$$\beta_{\text{difference}_i} \sim \text{Normal}(0, \sigma_\beta),$$

$$\sigma_\beta \sim \text{Cauchy}(0, 5)$$

Here, as suggested, the coefficient on the utilitarian component was set to be $(1-\beta_i)$. We used the negative variance as a parameter with coefficient β_i . We used a non-informative prior distribution for β as we did in model A. If $\beta_i = 0$, participants make completely utilitarian choices. If $\beta_i = 1$, participants make completely non-utilitarian choices.

First, we checked whether β_i (estimated by model B-2) and α_i (estimated by model A) were meaningfully correlated, as we had reported with model B in the old manuscript. As shown in the scatterplot below, α and β (this time from model B-2) were highly correlated, $r = 0.98$ (95% CI [0.97, 0.99]) as in the old manuscript.

Having confirmed that the model B-2 was fitted correctly, we quantified the evidence for model A over model B-2 using the Bayes factor. Calculated Bayes factor was $BF_{AB-2} > 1000$, indicating that model A is much more plausible than model B-2 for explaining the current data.

We have reported these results in the supplement:

Revised manuscript (lines 300-302):

“For more detailed model descriptions (including the specification of prior distributions, sensitivity analyses under different prior distributions, and a different model specification for mean-variance model), see the electronic supplementary material.”

(4) Were any precautions taken, or manipulation checks conducted, to make sure subjects actually believed that their decisions would affect money allocation to real people? I worry that social desirability bias could overwhelm subjects’ genuine moral intuitions if they correctly recognize that their allocations are fictional.

Response

As we described in the old manuscript (line 151), participants' allocations were real (Hsu et al., 2008; Kameda et al., 2016). Because we never use deception in experiments conducted by our laboratory, we think that our participants correctly recognize that their allocations would be actualized. We did not conduct manipulation checks about their beliefs in the current study, because asking such questions might cause ungrounded suspicion among participants that we use deception in experiments.

MINOR COMMENTS

(5) *The Introduction should more clearly distinguish between contributions of resource allocation parameters to actual choices versus to cognitive dynamics before the choice is made.*

Response

Thank you for the valuable suggestion. We have modified the introduction as follows:

Revised manuscript (lines 113-133)

“To test H2, we compared two models at the behavioural (choice) level. The first model (later described as model A) focuses on the worst-off elements of choice problems (i.e., the difference in minimum value between the two options) to explain participants' choice patterns. The second model (model B) focuses on the variance elements of choice problems (i.e., the difference in variance) to explain participants' choice patterns. As implied in H2, we predicted that model A would provide a better fit to the participants' choices than model B.

At the cognitive level, we predicted that compared to the variance elements, the worst-off elements would exert stronger influence on how straightforwardly participants move the mouse cursor to make choices. As illustrated in Figures 1b & 1c, the cursor's x -position was defined to have a larger positive value when the cursor approached the non-utilitarian options. H2 predicted that the cursor's x -position would be determined more strongly by the difference in minimum value than in variance. That is, the larger difference in minimum (rather than in variance) would make the trajectories of the cursor more straightforward to the non-utilitarian option.

Thirdly, for H3, we introduce a time course analysis of mouse trajectories to shed light on cognitive processes at a finer level. As seen in Figures 1b & 1c, the cursor's x -positions did not differ much initially, but diverge gradually between the two options in each time step. To capture how cognitive focus on the minimums over the variances (as claimed in H2) may temporally evolve during decision making, we need to assess the time courses of participants' mouse movements in each time step."

(6) Page 7, H2: Unclear what is meant by "parametric analysis". This is a vast class of statistical methods.

Response

As suggested, we have removed the phrase "parametric analysis":

Old manuscript (lines 109-111):

"parametric analysis will reveal that the worst-off elements will affect behavioural choices as well as cognitive (mouse tracking) processes more robustly, compared to the variance in allocated resources."

Revised manuscript (lines 109-111):

"the worst-off elements will affect behavioural choices as well as cognitive (mouse tracking) processes more robustly, compared to the variance in allocated resources."

(7) The Methods section confused with me with regard to statistical methods, for example referring to MCMC methods for "parameter estimation" before I had learned what the parameters of interest and models were. The later material showing the models and notation is good; please move it to Methods.

Response

According to the suggestion, we have moved the notation for models A & B and the state-space model to the "Materials and Methods" section in the revised manuscript.

(8) Page 12: “Parametric absolute differences” sounds like it refers to the estimates of a parametric model, but are you not just referring to differences between the fixed design parameters of the experiment?

Response

This is correct. As suggested, we have removed the word “parametric” from the sentence and simply described them as “absolute differences” (lines 226, 228).

(9) In the time-series analysis, are the confidence intervals simultaneous or pointwise? If pointwise, please also provide simultaneous CIs in the figures.

Response

We had reported pointwise confidence intervals. As suggested, we have added simultaneous confidence intervals:

The figure caption now reads:

Old manuscript (line 634):

“Shaded areas indicate a 95% highest density interval of estimated coefficients.”

Revised manuscript (lines 732-734):

“Shaded areas indicate 95% pointwise highest density interval of estimated coefficients. Dotted lines indicate 95% simultaneous highest density interval of estimated coefficients.”

(10) *Top of page 10: What is the scale of the estimated beta? Odds?*

Response

The scale of the estimated β is raw scale. Its range is unconstrained (it takes values from negative infinity to positive infinity). A smaller β indicates that a participant prefers utilitarian options, and a larger β indicates that a participant prefers non-utilitarian options. For actual range of estimated β , please see Figure 2b. We have clarified that the range of β is unconstrained as follows:

Old manuscript (lines 204-205):

“ β_i indicates the degree of the participant’s concern for the egalitarian (variance) parameter”

Revised manuscript (lines 195-198):

“ β_i is raw-scale and unconstrained (i.e., it ranges from negative infinity to positive infinity) and indicates the degree of the participant’s concern for the egalitarian (variance) parameter. A smaller β_i indicates that a participant prefers utilitarian options more, and a larger β_i indicates that a participant prefers non-utilitarian options more.”

(11) *Figure 3b: Around $t=30-40$, the coefficients for maximin and variance options are in different directions. Does this mean that during this time period, a higher variance parameter predicted mouse movement toward the *utilitarian* option? Why might this be? If I am interpretating this counterintuitive finding correctly, note that this is the kind of finding that could potentially arise spuriously due to aliasing.*

Response

Yes, around $t = 30 \sim 40$, a higher variance parameter predicted mouse movement toward the utilitarian option. As suggested, we have added caution

that this interesting (but difficult to interpret) result might have arisen from aliasing as follows:

Revised manuscript (lines 392-394):

“For example, we observed that around $t = 30 \sim 40$, a higher variance parameter predicted mouse movement toward the utilitarian option. This pattern is difficult to interpret but might have potentially arisen spuriously due to aliasing.”

(12) *Figure 1B: Hard to interpret this when the points aren't connected. Maybe take a random sample of the trajectories and plot them as lines in different colors.*

Response

As suggested, we have modified Figure 1b so that readers can interpret the trajectories easily:

The figure caption now reads:

Revised manuscript (lines 705-711):

“An illustration of eight mouse trajectories from one participant. Here, trajectories toward the left represent movements to utilitarian choices and the right represent movements to non-utilitarian choices. The initial location of the mouse cursor on the Next button was coded (0,0), the coordinate clicked to select the left option was coded (-1,1), and the coordinate clicked to select the right option was coded (1,1) in analysis. In the analysis below, we reset the x -position of the cursor to have a larger positive value as it approached a non-utilitarian option.”

(13) *The FigShare link does not seem to work.*

Response

We have fixed the broken link as follows:

Old manuscript (line 367):

<https://doi.org/10.6084/m9.figshare.12585482>

Revised manuscript (line 427):

<https://figshare.com/s/c501dc92b15a13452b66>

TYPOS

(14) *Page 5: “reveal real the time”*

Response

We have modified “reveal real the time” to “reveal the real-time” (line 83).

References

- Collins, L. M., Dziak, J. J., & Li, R. (2009). Design of experiments with multiple independent variables: a resource management perspective on complete and reduced factorial designs. *Psychological methods, 14*, 202–224.
- Hsu, M., Anen, C., & Quartz, S. R. (2008). The right and the good: Distributive justice and neural encoding of equity and efficiency. *Science, 320*, 1092–1095.

- Kameda, T., Inukai, K., Higuchi, S., Ogawa, A., Kim, H., Matsuda, T., & Sakagami, M. (2016). Rawlsian maximin rule operates as a common cognitive anchor in distributive justice and risky decisions. *Proceedings of the National Academy of Sciences of the United States of America*, *113*, 11817–11822.
- Nastase, S. A., Goldstein, A., & Hasson, U. (2020). Keep it real: rethinking the primacy of experimental control in cognitive neuroscience. *Neuroimage*, *222*, 117254.

Responses to the comments of Reviewer #3

(1) *The authors are not explicit enough about how their hypotheses are expected to be assessed in the data. This is particularly problematic for hypothesis 2 and 3, which established that the worse-off elements (minimum) should influence more strongly participants' choices than variance in allocated resources. From the moment in which these hypotheses are discussed (p.7), the reader should have an intuitive understanding of what exactly should they expect to see in mouse trajectories. This is not provided in the manuscript. As a result, it becomes hard to understand exactly why are they using the mouse-tracking methodology (as opposed to other time-sensitive measures, such as response times!) as well as why are they performing the analysis they do later on (time-step model). There are a number of possible things the authors could do to solve this issue. I mention two of them:*

a. The linking hypothesis between the hypothesised cognitive process (i.e. the variables that play a role in decision) and the mouse-trajectories should be made explicit. For example, the authors might expect that how fast or how straightforwardly participants make their non-utilitarian decision is determined by the difference in minimum values rather than by the difference in variance. If this were the case, one might even be able to notice a visual difference when comparing by-participant mean trajectories for trials where the difference between the two minima values is maximal and cases in which this difference is minimal. The predictions, stated in these lines, should be part of the paper.

b. The inclusion of figures showing the actual mouse tracking results might help the reader. For example, Figure 1b as it is not informative, as it doesn't have information about which trajectories correspond to utilitarian or non-utilitarian choices. Having a colour coding-scheme would be optimal. Similarly, it would be useful to have a figure illustrating the mean trajectories (or just the x-trajectory) for each kind of choice. That is, roughly illustrating the raw properties of mouse trajectories which are later used for the model.

Response

Thank you for the valuable suggestion. As explained below, we have (1) added a new figure and (2) modified the existing figure to help readers understand how our hypotheses 2 and 3 would be assessed with the mouse trajectory data. Additionally, based on these revised figures, (3) we have provided more

detailed explanation for our hypothesis that the difference in minimums (rather than the difference in variances) would determine how straightforwardly participants move the mouse cursor to make choices.

(1) First, as recommended, we have added a new figure (Figure 1c) to show an example participant's mean trajectories for utilitarian and non-utilitarian choices with separate color coding-scheme:

In the figure caption, we have added the following description:

Revised manuscript (lines 711-714):

“Each trajectory shows the example participant's mean trajectories for utilitarian (purple) and non-utilitarian (orange) choices. Shaded areas indicate a standard error of the mean of x -position at each time point (from $t = 1$ to $t = 101$).”

(2) Second, we have modified Figure 1b (see also Reviewer 2's comment 12). We took a random sample of example trajectories and plotted them as differently colored lines:

The figure caption now reads:

Revised manuscript (lines 705-711):

“An illustration of eight mouse trajectories from one participant. Here, trajectories toward the left represent movements to utilitarian choices and the right represent movements to non-utilitarian choices. The initial location of the mouse cursor on the Next button was coded (0,0), the coordinate clicked to select the left option was coded (-1,1), and the coordinate clicked to select the right option was coded (1,1) in analysis. In the analysis below, we reset the x -position of the cursor to have a larger positive value as it approached a non-utilitarian option.”

(3) Finally, to explicitly link the hypothesized cognitive process (i.e. the variables that play a role in decision) and the mouse trajectories, we have modified the text as follows:

Revised manuscript (lines 120-133):

“At the cognitive level, we predicted that compared to the variance elements, the worst-off elements would exert stronger influence on how straightforwardly participants move the mouse cursor to make choices. As illustrated in Figures 1b & 1c, the cursor’s x -position was defined to have a

larger positive value when the cursor approached the non-utilitarian options. H2 predicted that the cursor's x -position would be determined more strongly by the difference in minimum value more than those in variance. That is, the larger difference in minimum (rather than in variance) would make the trajectories of the cursor more straightforward to the non-utilitarian option. Thirdly, for H3, we introduce a time course analysis of mouse trajectories to shed light on cognitive processes at a finer level. As seen in Figures 1b & 1c, the cursor's x -positions did not differ much initially, but diverged gradually between the two options in each time step. To capture how cognitive focus on the minimums over the variance (as claimed in H2) may temporally evolve during decision making, we need to assess the time courses of participants' mouse movements in each time step."

We believe that these additional descriptions would help readers gain an intuitive understanding of what they should expect to see in mouse trajectories and why we analyzed the mouse-tracking data in each time step using a time-series model.

(2) In relation with the first point, I find the description of the time-series analysis performed on mouse trajectories not fully transparent. I think the authors should provide a more complete explanation of what each of the terms is doing, and how exactly are we supposed to interpret the figure 3b. This could be done in the Supplementary Materials, but note that, as it is, the formula in page 13 are not useful, as it's unclear exactly how they are extracted and what exactly each term means (for instance, is there a difference between η_{min} and η_{min} ?)

Response

We regret the inadequate explanation of the time-series model in the old manuscript. In fact, there is no difference between η_{min} and η_{min} , so we have aligned the formula to unify η_{min} to η_{min} in the revised manuscript (line 245). We have also added detailed descriptions of each term as Table S2 in the supplement:

Detailed descriptions of time-series model.

Parameter	Description
-----------	-------------

$\mu_{t,i}$	Intercept for participant i at time t
$\beta_{Min_{t,i}}$	Coefficient for the absolute difference of minimums for participant i at time t
$\beta_{Var_{t,i}}$	Coefficient for the absolute difference of variances for participant i at time t
ε	Observation error in the state space modeling
δ_i	Process error for the intercept in the state space modeling
$\beta_{Min_population_t}$	Population level coefficient for the absolute difference of minimums at time t
η_{Min_t}	Individual differences in $\beta_{Min_{t,i}}$ at time t
$\beta_{Var_population_t}$	Population level coefficient for the absolute difference of variances at time t
η_{Var_t}	Individual differences in $\beta_{Var_{t,i}}$ at time t
ζ_{Min}	Process error for $\beta_{Min_population_t}$ in the state space modeling
ζ_{Var}	Process error for $\beta_{Var_population_t}$ in the state space modeling
σ_{δ_i}	Scale (standard deviation) parameter for δ_i
$\sigma_{\eta_{Min_t}}$	Scale parameter for η_{Min_t}
$\sigma_{\eta_{Var_t}}$	Scale parameter for η_{Var_t}
$\sigma_{\zeta_{Min}}$	Scale parameter for ζ_{Min}
$\sigma_{\zeta_{Var}}$	Scale parameter for ζ_{Var}

We have also added a more detailed explanation about how to interpret Figure 3b in its caption:

Revised manuscript (lines 729-732):

“Here, $\beta_{Min_population_t}$ quantified the extent to which the absolute differences in minimums predicted mouse movement toward the non-utilitarian option at time t . Around $t = 1$ to $t = 30$, $\beta_{Min_population_t}$ was not meaningfully different from zero (a horizontal line).”

(3) *The authors need to clarify that what is meant by “decision time” is the time when participants click on the response button (i.e. response time). This is not obvious in MT*

literature, where people use the method precisely to determine the exact moment in which participants make the decision, which happens before the actual click.

Response

Thank you for the important comment. As suggested, we have replaced “decision time” with “response time” in the revised manuscript (lines 306, 307 and Figure 3a) and added a definition of response time as follows:

Revised manuscript (lines 305-307):

“Figure 3a displays a distribution of participants’ response times that elapsed from the trial onset to their clicking one of the choice buttons.”

(4) The description of the design of the experiment is rather confusing. The authors present the picture as if they are manipulating whether the choices are utilitarian vs. Not-utilitarian, but I think they could make clearer how these options relate to the manipulation of Total, Variance and Minimum (they do an attempt of this I p.88 lines 143-148, but it’s not explicitly presented as actual factors they are manipulating).

Response

We regret the confusing description in the previous manuscript. We have clarified this point by adding a detailed description to the note of Table S1 about how the 48 choice sets were constructed:

Revised manuscript:

“These 48 choice sets were constructed to distinguish two types of distributive concerns in choices [i.e., utilitarian (larger total) and non-utilitarian (smaller variance and larger minimum)]. Out of the 48 choice sets, the egalitarian and utilitarian parameters are orthogonal to each other in 24 choice sets (i.e., the Pearson correlation coefficient between absolute differences in variance and that in total is zero for these 24 choice sets), and the maximin and utilitarian parameters are orthogonal to each other in the remaining 24 choice sets (i.e., the Pearson correlation coefficient between absolute differences in minimum and that in total is zero).”

(5) *What is figure 3a trying to show? Wouldn't be more pertinent to have an idea about whether the manipulation has some effect on decision time? (See also my point above about the meaning of decision time)*

Response

Thank you for the valuable suggestion. As we believe that presenting the response time distribution would help readers understand the time scale of our mouse-tracking analysis, we have decided to keep Figure 3a. However, as suggested, we have modified the text in the figure and replaced “decision time” with “response time”:

(6) *It's not always clear whether the analysis of mouse-tracking data is being made only for trials where the final choice was non-utilitarian or for all trials. I guess it's always the former, but clarification is needed.*

Response

Thank you for raising this important point. We regret the unclear explanation of the analysis of mouse-tracking data in the old manuscript. We used the data from all the trials regardless of the final choice (see Sullivan et al., 2015). We have clarified this point as follows:

Revised manuscript (lines 277-279):

“In all analyses including the mouse-tracking analysis, we used all the data from each trial, whether participants chose the utilitarian or the non-utilitarian option in the trial.”

Additional Comments about statistics

(7) *In section “Estimation” (p.9, line 174), it’s unclear what model are they referring to when they talk about “the model”, as well as what is the R^{\wedge} statistics they have run... (is it a correlation?). I believe the authors can chose to be vague in the paper about the exact statistics they did if and only if they provide all the details in the supplementary materials, which is not the case. One should be able to run the analyses on their data, and understand exactly what they did.*

Response

We regret the vague explanation in Estimation section. By saying “the model”, we referred to all the models we used in the manuscript, including the mixed-effects logistic regression in section “choice data”, models A and B in section “Model-based choice analysis,” and the state-space model in section “Mouse tracking data”. To clarify which models we are referring to in the “Estimation” section, we have moved the notation of models A, B, and the state-space model to the front. These notations have been placed in “Materials and Methods” in the revised manuscript (instead of “Results” as in the old manuscript).

Old manuscript (line 175):

“The model was implemented using rstan version 2.19.3”

Revised manuscript (lines 270-272):

“All the models we used in the current study, including models A and B, the logistic regression (to be mentioned later), and the state-space model, were implemented using rstan version 2.19.3”

We should have also explained what the \hat{R} statistic (the Gelman-Rubin convergence statistic) means in the old manuscript. The \hat{R} statistics are used for checking the convergence of parameter estimations by MCMC methods (Kruschke, 2015). If the \hat{R} statistics are below 1.1, it is commonly assumed that the estimation has finished successfully (i.e., the Markov chains for each

parameter converged to a stationary distribution; Gelman et al., 2014). We have described what the \hat{R} statistics indicate in the revised manuscript:

Old manuscript (line 177):

“The \hat{R} statistics were below 1.1 for all parameters.”

Revised manuscript (lines 273-276):

“We used the \hat{R} statistic (the Gelman-Rubin convergence statistic) to check for convergence of all models’ parameter estimations [32]. The \hat{R} statistics were below 1.1 for all the parameters we estimated using MCMC in the current study, indicating the convergence of our MCMC simulations.”

(8) Bayes factors are generally reported together with some measure of their robustness under different prior specifications. This is lacking (as well as any reference to which prior specification was used).

Response

Thank you for the valuable suggestion. We should have clearly stated that we had checked the robustness of our findings about Bayes factors in the old manuscript. Please see our response to this point raised in Reviewer 2’s Comment #2.

References

- Gelman, A., Carlin, J. B., Stern, H. S., Dunson, D. B., Vehtari, A., and Rubin, D. B. (2013). *Bayesian data analysis*. Chapman and Hall/CRC
- Kruschke, J. K. (2015). *Doing Bayesian data analysis: A tutorial with R, JAGS, and Stan*. Academic Press.
- Sullivan, N., Hutcherson, C., Harris, A., & Rangel, A. (2015). Dietary self-control is related to the speed with which attributes of healthfulness and tastiness are processed. *Psychological science*, 26(2), 122–134.
<https://doi.org/10.1177/0956797614559543>

We are greatly indebted to the editor and the three reviewers for the thoughtful comments on our manuscript. Thanks to them, we believe that the new

manuscript is greatly improved in transparency of procedure and results. We hope that you will agree.

Sincerely yours,

Tatsuya Kameda, Ph.D.
Professor
Department of Social Psychology
The University of Tokyo
Tokyo, Japan

Appendix B

January 22, 2021

Dear Dr. Essi Viding:

Thank you for your and your reviewers' thoughtful comments on our manuscript "Reducing variance or helping the poorest? A mouse tracking approach to investigate cognitive bases of inequality aversion in resource allocation" submitted for publication in *Royal Society Open Science* (RSOS-201159). We are deeply grateful to the editor for giving us the opportunity to revise and resubmit. We have taken the reviewers' comments into careful consideration and revised our manuscript to reflect them. Below we summarize how we have responded to each of the points raised by the two reviewers. The experiment, figure, table, and line numbers refer to those in the revised manuscript unless otherwise specified.

Responses to the comments of Reviewer #2

(1) *I appreciate the authors' clarification of how the choice sets were designed, but this structure of choice sets is a more serious problem for the central conclusions of the manuscript than the authors' revision suggests.*

It is fine to manipulate parameters in an aliased manner in order to address an ecological hypothesis, as the revision implies, but then one simply cannot rigorously address hypotheses such as H2, one of the paper's two central hypotheses ("...choices and mouse trajectories by the minimum parameter than by the egalitarian (variance) parameter"), at least with the current analysis approach. Accordingly, central conclusions appearing throughout the Abstract and main text, such as:

"Our results provide the first...evidence that people weight the maximin concern over the egalitarian concern..."

could simply be spurious. It could simply be that other (confounded) aspects of the choice sets, such as the total, is driving the apparent evidence in favor of subjects' weighting maximin concerns over egalitarian concerns.

There are statistical methods to address this analytically, e.g., by doing subset analyses in which the variables of interest (e.g., minimum and variance) are not confounded by other manipulated elements (e.g., the total) or *by statistically controlling for the other manipulated elements*. These are methods that control for confounding, because aliasing is a type of confounding. I would strongly encourage the authors to look into how they could resolve this problem rigorously by considering the specific structure and resolution of the choice sets. *It may be as simple as revising all models to include all manipulated choice set parameters rather than just the ones of interest (thus controlling for the other, confounded parameters)*.

Response

Thank you for raising this critical point. We regret that we missed what the reviewer pointed out about the mouse tracking analysis in the previous revision. Following the reviewer's suggestion [i.e., *revising all models to include all manipulated choice set parameters rather than just the ones of interest (thus controlling for the other, confounded parameters)*], we have conducted an additional analysis by statistically controlling for the total element (i.e., the other parameter manipulated in the choice sets) in Eq. 3:

$$x_{t,i,j} = \mu_{t,i} + \beta_{Min_{t,i}} \times Diff_min_j + \beta_{Var_{t,i}} \times Diff_var_j$$

$$+ \beta_{Total_{t,i}} \times Diff_total_j + \varepsilon,$$

$$\mu_{t,i} = \mu_{t-1,i} + \delta_i,$$

$$\beta_{Min_{t,i}} = \beta_{Min_population_t} + \eta_{Min_t},$$

$$\beta_{Var_{t,i}} = \beta_{Var_population_t} + \eta_{Var_t},$$

$$\beta_{Total_{t,i}} = \beta_{Total_population_t} + \eta_{Total_t},$$

$$\beta_{Min_population_t} = \beta_{Min_population_{t-1}} + \zeta_{Min},$$

$$\beta_{Var_population_t} = \beta_{Var_population_{t-1}} + \zeta_{Var},$$

$$\beta_{Total_population_t} = \beta_{Total_population_{t-1}} + \zeta_{Total},$$

where $\varepsilon \sim N(0, 0.005)$, $\delta_i \sim N(0, \sigma_{\delta_i})$, $\eta_{Min_t} \sim N(0, \sigma_{\eta_{Min_t}})$, $\eta_{Var_t} \sim N(0, \sigma_{\eta_{Var_t}})$,

$$\eta_{Total_t} \sim N(0, \sigma_{\eta_{Total_t}}), \zeta_{Min} \sim N(0, \sigma_{\zeta_{Min}}),$$

$$\zeta_{Var} \sim N(0, \sigma_{\zeta_{Var}}), \text{ and } \zeta_{Total} \sim N(0, \sigma_{\zeta_{Total}}).$$

Below, the left figure shows the results from the analysis reported in the main text (i.e., Figure 3b), and the right figure shows the results from the additional analysis.

As shown in the graph, after controlling for $\beta_{Total_population_t}$, we obtained very similar results for both $\beta_{Min_population_t}$ and $\beta_{Var_population_t}$. Although the potential problem of aliasing due to the moderate correlation between the maximin and variance parameters is not completely solved (as acknowledged in the discussion), we believe that this additional analysis helps ameliorate the statistical problem in the revised manuscript, thanks to the reviewer's useful suggestion. We have reported the additional analysis in the supplement and included a description of the analysis in the revised manuscript as follows:

Revised manuscript (lines 337-340):

"We also conducted an additional analysis by statistically controlling for the total element in the mouse tracking analysis and confirmed that results were essentially unchanged. See the electronic supplementary material ('*Mouse tracking analysis after controlling for the total element*') and Table S4."

(2) The sensitivity analyses regarding the prior on beta are a good, and reassuring, addition. Thank you.

However, with alpha, you state that its $U(0,1)$ is noninformative, but can't alpha take on negative values as well, at least in principle? Bounding alpha above 0 excludes the possibility that any subject wants the minimum to be *smaller* rather than larger. We don't expect that to be the case on average, but this prior entirely excludes this possibility for every individual subject, which would seem to be a rather strong assumption. It is rarely advisable to use priors that completely exclude some parts of the parameter space. I would advise performing a sensitivity analysis using a genuinely noninformative or highly diffuse prior, similar to what was done for beta in the revision.

Response

As suggested, we have modified model A so that the alpha (maximin parameter) could take negative values as well. Specifically, we have defined model A-2 based on the specification of model B to calculate Bayes factor with the same prior settings. Here, model A-2 is given by:

$$U_i(x) = \frac{1}{3}(\pi_1 + \pi_2 + \pi_3) - \alpha_i \times (-\min[\pi_1, \pi_2, \pi_3]),$$

$$\alpha_i = \alpha_{population} + \alpha_{difference_i},$$

$$\alpha_{population} \sim Normal(\text{mean}, \text{sd}),$$

$$\alpha_{difference_i} \sim Normal(0, \sigma_\alpha),$$

$$\sigma_\alpha \sim Cauchy(0, 5)$$

According to the reviewer's suggestion, we have calculated Bayes factor (model A-2 vs. B) with three different prior settings for $\alpha_{population}$ and $\beta_{population}$ in model A-2 and B, respectively. Specifically, we have calculated Bayes factor under the prior distributions of (1) mean = 5 and sd = 15, (2) mean = 0 and sd = 15, and (3) mean = 0 and sd = 100 for $\alpha_{population}$ and $\beta_{population}$.

	Mean = 5, sd = 15	Mean = 0, sd = 15	Mean = 0, sd = 100
Bayes factor (model A-2 vs. B)	BF > 100	BF > 100	BF > 100

As seen in the table above, model A-2 is better than model B in terms of Bayes factors regardless of the prior distributions. Accordingly, we can safely assume that models including minimum parameters (i.e., alpha) are better than those including egalitarian parameters (i.e., beta), even when allowing the alpha to take negative values.

We have also checked whether α_i (estimated by model A-2) and β_i (estimated by model B) were meaningfully correlated across different prior settings to verify that our estimation has been correctly performed. Please note that we standardized minimums in model A-2 and variances in model B when we fitted these models. As shown in the scatterplots below, α (from model A-2) and β (from model B) were highly correlated.

We have reported these results in the supplement and referenced them in the revised manuscript as follows:

Revised manuscript (lines 307-310):

“For more detailed model descriptions (including the specification of prior distributions, sensitivity analyses under different prior distributions, and different model specifications for mean-variance model and quasi-maximin model), see the electronic supplementary material.”

(4) My concern was about subject's beliefs about the veracity of their allocations, not about the veracity of the allocations themselves. I would expect at least some mention of this limitation, given that no manipulation checks were performed. (Minor point: regarding the actual veracity of the allocation, the old and revised manuscript just say "The participants were paid after the experiment" at the indicated line; you might revise this indicate that they were actually paid according to the allocations).

Response

We apologize for our misunderstanding in the previous manuscript. As suggested, we have added the following descriptions to the text:

Revised manuscript (lines 172-173):

"The recipients were paid according to the allocations after the experiment."

Revised manuscript (lines 394-398):

"Thirdly, we did not conduct manipulation checks to confirm that participants actually believed that their decisions would affect monetary allocation to real people. We did not conduct manipulation checks because asking such questions might cause ungrounded suspicion among participants that we use deception in our experiments. However, the lack of belief checks remains as a limitation of the current study."

Responses to the comments of Reviewer #3

(1) One of my suggestions in my previous review was to include a figure showing mean mouse tracking trajectories per type of response. In particular, I think it would be very helpful to see mean trajectories across subjects which include some by-participant measure of variability (e.g., SE on participant-means). In the new Fig.1c, the authors include mean trajectories but only for one participant. This decision is rather odd to me (i.e., how did they choose what participant to show), and it doesn't really target my original concern, which was having some figure that shows how much variability is in there (between subjects variability). Note that Fig1b is already there and does a great job in showing *_within_* subject variability. I think the authors should replace Fig1c with a new figure showing mean trajectories across subjects.

Response

Thank you for raising this important point. As suggested, we have revised Figure 1C so that it shows mean trajectories across participants with SE.

We have incorporated this figure as Figure 1C in the revised manuscript.

(2) The caption of Fig1 can also be improved. For example, each of three subcaptions should be unified (right now there are two a/b/c).

As suggested, we have modified the caption of Figure 1 as follows:

Previous manuscript (lines 694-715):

“An illustration of (a) a choice problem with two choice options (the left and right columns), (b) a participant’s mouse trajectories for 8 choice problems randomly sampled from the 48 choice problems, and (c) the same participant’s mean trajectories for utilitarian and non-utilitarian choices across the 48 choice problems. (a) Participants were told that person A would receive the lowest amount (under the “Minimum” label), person B the middle (“Medium”), and person C the highest amount (“High”) in the chosen option (all in Yen). They were also instructed that numbers under the “Variance” label represented Gini coefficients used in economics that could range from 0 (perfect equality) to 1 (perfect inequality), and those under the “Total” label indicated the sum of the allocation amounts to the three recipients. In this example, the left is the Utilitarian option, and the right is the non-Utilitarian option. In each trial, the mouse cursor was initially at the “Next” button, which became clickable after participants had indicated their choice (left or right) using the button located at the top of the screen. (b) An illustration of eight mouse trajectories from one participant. Here, trajectories toward the left represent movements to utilitarian choices and the right represent movements to non-utilitarian choices. The initial location of the mouse cursor on the Next button was coded (0,0), the coordinate clicked to select the left option was coded (-1,1), and the coordinate clicked to select the right option was coded (1,1) in analysis. In the analysis below, we reset the x -position of the cursor to have a larger positive value as it approached a non-utilitarian option. (c) Each trajectory shows the example participant’s mean trajectories for utilitarian (purple) and non-utilitarian (orange) choices. Shaded areas indicate a standard error of the mean of x -position at each time point (from $t = 1$ to $t = 101$).”

Revised manuscript (lines 709-728):

“Figure 1. Illustrations of a choice problem with two alternatives (the left and right columns) and participants’ mouse trajectories. (a) Participants were told that person A would receive the lowest amount (under the “Minimum” label), person B the middle (“Medium”), and person C the highest amount (“High”) in the chosen option (all in Yen). They were also instructed that numbers under

the “Variance” label represented Gini coefficients used in economics that could range from 0 (perfect equality) to 1 (perfect inequality), and those under the “Total” label indicated the sum of the allocation amounts to the three recipients. In this example, the left is the Utilitarian option, and the right is the non-Utilitarian option. In each trial, the mouse cursor was initially at the “Next” button, which became clickable after participants had indicated their choice (left or right) using the button located at the top of the screen. (b) An illustration of eight mouse trajectories from one participant. Here, trajectories toward the left represent movements to utilitarian choices and the right represent movements to non-utilitarian choices. The initial location of the mouse cursor on the Next button was coded (0,0), the coordinate clicked to select the left option was coded (-1,1), and the coordinate clicked to select the right option was coded (1,1) in analysis. In the analysis below, the non-utilitarian response was always analysed as being located at $x = 1$ (i.e., the x -position of the cursor had a larger positive value as it approached a non-utilitarian option). (c) Mean trajectories across all participants for utilitarian (purple) and non-utilitarian (orange) choices. Shaded areas indicate a standard error of the mean of x -position at each time point (from $t = 1$ to $t = 101$).”

(3) *In the caption of figure 1b, the authors say that "In the analysis below, we reset the x -position of the cursor...". If I understand correctly, what the authors mean by this is that the non-utilitarian answer will always be analysed as being located at $x=1$. If this is the case, this is not clear from their description.*

We have modified the caption of Figure 1B as suggested. Please see our response to the comment above (Comment #2).

(4) *At the point in which H2 is introduced (lines 110-114), it's still unclear for the reader how the manipulation of the minimum parameter and the egalitarian variance is done. I think it will be helpful to, using the example in Fig.1a, to clearly explain that in a trial where the minimum difference is larger between the two options, people are expected to go more straightforward to the non-utilitarian option than in a trial where the main difference between the options regards the variance.*

Response

As suggested, we have added an explanation as follows:

Revised manuscript (lines 98-102):

“In Figure 1a, for example, the absolute difference in minimum (maximin parameter) is 420 and that in variance (egalitarian parameter) is 0.2. These differences varied from trial to trial. In a trial where the minimum difference is larger between the two options, people are expected to go more directly to the non-utilitarian option than in a trial where the main difference between the options regards the variance.”

We are greatly indebted to the editor and the two reviewers for the thoughtful comments on our manuscript. Thanks to them, we believe that the new manuscript is greatly improved in transparency of procedure and results. We hope that you will agree.

Sincerely yours,

Tatsuya Kameda, Ph.D.
Professor
Department of Social Psychology
The University of Tokyo
Tokyo, Japan